# Federated Learning on Adaptively Weighted Nodes by Bilevel Optimization

## Abstract

We propose a federated learning method with weighted nodes in which the weights can be modified to optimize the model's performance on a separate validation set. The problem is formulated as a bilevel optimization problem where the inner problem is a federated learning problem with weighted nodes and the outer problem focuses on optimizing the weights based on the validation performance of the model returned from the inner problem. A communication-efficient federated optimization algorithm is designed to solve this bilevel optimization problem. We analyze the generalization performance of the output model and identify the scenarios when our method is in theory superior to training a model locally and superior to federated learning with static and evenly distributed weights.

## 1 Introduction

Federated learning (FL) is an emerging technique for training a model using data distributed over a network of nodes without sharing data between nodes (Konečný et al., 2016; McMahan et al., 2017). In this paper, we focus on the case where data distributions across nodes are heterogeneous and each node aims at a model with an optimal local generalization performance. In the classical setting of FL, a globally shared model is learned by minimizing a weighted average loss across all nodes. However, given the heterogeneity of data distributions, a global model is likely to be sub-optimal for some node (Fallah et al., 2020). Alternatively, each node can train a model only using its local data, but such a local model may not generalize well neither when the volume of local data is small.

To achieve a good local generalization performance, each node can still exploit global training data through FL but, at the same time, identify and collaborate only with the nodes whose data distributions are similar or identical to its local distribution. One way to implement this strategy is to allow each node to solve its own weighted average loss minimization problem with weights designed based on the performance on a separate set of local (validation) data. Ideally, each node can learn a better model by allocating more weights on its peers whose data distribution is similar to its local distribution. In this paper, we formulate the choice of the weights as a bilevel optimization (BO) problem (Colson et al., 2005; Vicente & Calamai, 1994), which can be solved by a federated bilevel optimization algorithm, and analyze the generalization performances of the resulting model.

We consider a standard learning problem where the goal is to learn a vector of model parameters $\theta$ from a set $\Theta$ that minimizes a generalization loss. This problem can be formulated as

$$\theta^* \in \arg\min_{\theta \in \Theta} \left\{ L_0(\theta) := \mathbb{E}_{z \sim p_0}\left[ l(\theta; z) \right] \right\}, \tag{P}$$

where $l(\theta; z)$ is the loss of $\theta$ on a data point $z$ from a space $\mathcal{Z}$, and $\mathbb{E}_{z \sim p_0}$ represents the expectation taken over $z$ when $z$ follows an unknown ground truth distribution $p_0$.

Directly solving (P) is challenging as $p_0$ is unknown, and, typically, training data sampled from $p_0$ is needed for learning an approximation of $\theta^*$. In this paper, we consider the scenario where the amount of data sampled directly from $p_0$ may not be sufficient to learn a good approximation of $\theta^*$, but there exist external data distributed on $K$ nodes that can potentially help the learning on $\theta^*$. In particular, we denote the set of nodes by $\mathcal{K} := \{1, \dots, K\}$ and assume a training set $D_k^{\text{train}}$ is stored in node $k$. We also define $D^{\text{train}} := \left\{ D_k^{\text{train}} \right\}_{k=1}^K$ and assume $|D_k^{\text{train}}| = n_k$ and $D_k^{\text{train}} = \{z_k^{(i)}\}_{i=1}^{n_k}$, where $z_k^{(i)} \in \mathcal{Z}$ is an i.i.d. sample from an unknown distribution $p_k$ for $k \in \mathcal{K}$.

We assume node $k$ is weighted by $w_k$ and the vector of weights $w = (w_1, \ldots, w_K) \in [0, 1]^K$ is located on the capped simplex $\Delta_K^b$ defined as

$$\Delta_K^b = \left\{ w = (w_1, \ldots, w_K) \middle| \sum_{k=1}^K w_k = 1, 0 \le w_k \le b, k \in \mathcal{K} \right\},$$

where $b \in [\frac{1}{K}, 1]$ is a user-defined parameter. The FL on weighted nodes can be formulated as

$$\widehat{\theta}(w) \in \arg\min_{\theta \in \Theta} \sum_{k=1}^K w_k \widehat{L}_k(\theta), \tag{1}$$

where $\widehat{L}_k(\theta)$ is the empirical loss of $\theta$ on $D_k^{\text{train}}$, namely,

$$\widehat{L}_k(\theta) := \frac{1}{n_k} \sum_{i=1}^{n_k} l(\theta; z_k^{(i)}), \quad k \in \mathcal{K}. \tag{2}$$

When some $p_k$'s are different from $p_0$, $w$ in (1) must be chosen adaptively to ensure $\widehat{\theta}(w)$ is a good approximation of $\theta^*$ in (P). To do so, we assume that there is a validation dataset $D^{\text{valid}}$ with $|D^{\text{valid}}| = n_0 = n_{\text{valid}}$ and $D^{\text{valid}} = \{z^{(i)}\}_{i=1}^{n_0}$, where $z^{(i)} \in \mathcal{Z}$ is an i.i.d. sample from $p_0$. We assume $D^{\text{valid}}$ is stored in a node called node 0 or *center*, which may or may not be a node in $\mathcal{K}$. Set $D^{\text{valid}}$ alone may not be sufficient for learning $\theta^*$ precisely but can be used to assist the selection of $w$. We then propose to estimate the generalization loss of $\widehat{\theta}(w)$ using the loss on $D^{\text{valid}}$, i.e.,

$$\widehat{L}_0(\theta) := \frac{1}{n_0} \sum_{i=1}^{n_0} l(\theta; z^{(i)}) \tag{3}$$

and use this validation loss to guide the procedure for updating $w$. Presumably, when both the training and validation sets are large enough, the weights in $w$ will be shifted towards the nodes where the data is helpful for learning $\theta^*$. Following this idea, we formulate the *federated learning problem on adaptively weighted nodes* as the following *bilevel optimization* (BO) problem:

$$\widehat{w} \in \arg\min_{w \in \Delta_K^b} \left\{ \widehat{F}(w) := \widehat{L}_0(\widehat{\theta}(w)) \text{ s.t. } \widehat{\theta}(w) \text{ is defined as in (1)} \right\}. \tag{$\widehat{\text{P}}$}$$

In Section 4, we will present a federated optimization algorithm for solving ($\widehat{\text{P}}$). Suppose an algorithm can find the optimal solution $\widehat{w}$ of ($\widehat{\text{P}}$) and the corresponding model parameter $\widehat{\theta}(w)$. We are interested in the optimality gap of the generalization loss of $\widehat{\theta}(\widehat{w})$, namely, $L_0(\widehat{\theta}(\widehat{w})) - L_0(\theta^*)$, where $L_0$ is defined as in (P). The *main contribution of this paper* is to establish a high-probability bound of this gap as a function of the sizes of $D^{\text{train}}$ and $D^{\text{valid}}$ as well as a statistical distance between $p_0$ and $p_k$'s. Moreover, we compare our generalization bound with the bound achieved by learning only locally from $D^{\text{valid}}$ and the bound achieved by solving (1) with evenly distributed weights, and identify the parameter regimes where our method is preferred in theory.

## 2 RELATED WORK

The work most related to ours is Chen et al. (2021a) in which the authors proposed a target-aware weighted training algorithm for cross-task learning. Although their problem is completely different from FL, the bilevel optimization model they studied contains ($\widehat{\text{P}}$) as a special case. In fact, some steps in the proofs of the generalization bounds in the current work are borrowed from Chen et al. (2021a) with some modifications. However, our work extends their results in several valuable directions. First, the generalization bound in Chen et al. (2021a) is shown for any weight $w$ without any small or zero components, which is not necessarily the case for the optimal solution $\widehat{w}$ of ($\widehat{\text{P}}$). Second, their generalization bound contains a term of task distance whose convergence rate is not characterized. On the contrary, we show the convergence of the entire generalization bound for $\widehat{w}$ without any conditions on its components. Third, the generalization bound in Chen et al. (2021a) has a dominating term $O(1/\sqrt{n_{\text{valid}}})$, which is the same as the generalization bound obtained by directly training with the local data $D^{\text{valid}}$. However, we show that, when there exist identical neighbors and an error bound condition holds (Assumptions 2' and 3), the model learned by ($\widehat{\text{P}}$) can be superior to a model trained locally when the $p_k$'s are similar enough to (but still different from) $p_0$, providing an insight on when a node with insufficient data should actively seek collaboration with others.

FL has become a prominent machine learning paradigm for training models with distributed data Konečný et al. (2016); McMahan et al. (2017). Many federated optimization algorithms have been

developed for solving (1) or its expectation form (with $\widehat{L}_k$ replaced by $L_k$). A well-known method is the federated averaging (FedAvg) method (McMahan et al., 2017), which applies a local optimization method (e.g., stochastic gradient descend (Robbins & Monro, 1951)) to $\widehat{L}_k$ or $L_k$ in each node and periodically aggregates the solutions from all nodes by averaging. Many variants of FedAvg and other federated learning methods have been proposed to reduce the computation and communication complexity. A partial list includes Gorbunov et al. (2021); Lee et al. (2017); Karimireddy et al. (2020); Liang et al. (2019); Li et al. (2020); Yuan & Ma (2020); Wu & Wang (2021); Zhao et al. (2021). In our setting, (1) is a sub-problem we need to solve multiple times with different $w$'s. We then apply the Local-SVRG method by Gorbunov et al. (2021) to (1) because it has the lowest communication complexity for finite-sum problems like (1).

Most FL methods produce a globally-shared model which may not perform well on each node when data is heterogeneous across nodes. To address this challenge, many personalized FL methods, including but not limited to Smith et al. (2017); Tan et al. (2022); Fallah et al. (2020); Li & Wang (2019); Deng et al. (2020); Li et al. (2021), have been developed, where a global model is tailored using local data for a good local performance. However, many personalized FL methods use a fixed weight in (1) to obtain the global model. Such a global model may be dominated by the majority of the data distributions in the network and is hard to personalize for a minority group with unique data patterns. On the contrary, our method can produce a personalized weight so a node from the minority group can still find and collaborate with its peers.

BO recently has also been studied actively by the machine learning community. Many efficient optimization algorithms have been developed recently for BO, including but not limited to Chen et al. (2021c;b); Ghadimi & Wang (2018); Hong et al. (2020); Guo et al. (2021); Ji et al. (2021); Grazzi et al. (2020). However, these algorithms are designed for a single-machine setting and may not be communication efficient if implemented directly in a distributed environment. There are much fewer studies on BO in a distributed setting. The recent works (Li et al., 2022; Tarzanagh et al., 2022) consider a BO where both the outer and inner problems are defined with the expectation over data distributed across nodes. They analyze the communication complexity of their methods in a non-convex setting. We propose a different FL algorithm based on Local-SVRG because our problem $(\widehat{P})$ has a finite-sum structure that allows periodically going through all the data points in each node to obtain exact gradient information and achieving lower communication complexity than Li et al. (2022); Tarzanagh et al. (2022). (Chen et al., 2022) consider a decentralized BO problem and their algorithm for the deterministic case can be applied to our problem and achieve the same communication complexity in the non-convex case. However, we include the results for the convex case and focus more on the generalization performance of the federated learning based on BO.

## 3 GENERALIZATION PERFORMANCE

The following assumption on $(\widehat{P})$ is made for analyzing the generalization performance of $\widehat{\theta}(\widehat{w})$ in $(\widehat{P})$ and the convergence property of the optimization algorithm for solving $(\widehat{P})$ in Section 4.

**Assumption 1** (Well-behaved function). *The following statements hold. (1) $l(\theta; z) \in [0, 1]$ and $\nabla l(\theta; z)$ is $\ell_1$-Lipschitz continuous in $\theta$ for any $z \in \mathcal{Z}$. (2) $\widehat{L}_k(\theta)$ and $\nabla^2 \widehat{L}_k(\theta)$ are $\ell_0$ and $\ell_2$-Lipschitz continuous, respectively, for $k \in \mathcal{K}$. (3) $\widehat{L}_k(\theta)$ is $\mu$-strongly convex for $k \in \mathcal{K}$.*

These are standard regularity assumptions in recent literature on bilevel optimization (e.g. Ghadimi & Wang (2018)). Assuming the strong convexity in the lower-level problem, (1) has a unique solution so that the inclusion there can be replaced by equality. Similar to $L_0$ in (P), we define

$$L_k(\theta) := \mathbb{E}_{z \sim p_k}[l(\theta; z)] \text{ for } k \in \mathcal{K},$$

and we consider the following auxiliary problem

$$\mathcal{W}^* = \underset{w \in \Delta_K^b}{\arg\min} \left\{ F(w) := L_0(\theta(w)) \text{ s.t. } \theta(w) \in \underset{\theta \in \Theta}{\arg\min} \sum_{k=1}^K w_k L_k(\theta) \right\}. \tag{P_*}$$

Problem $(\widehat{P})$ can be viewed as an empirical approximation of $(P_*)$ in both inner and outer problems.

Even if all $p_k$'s are different from $p_0$, it is still possible to learn $\theta^*$ correctly by solving $(P_*)$. A simple example on mean estimation is $\min_{w \in \Delta_2^1} \mathbb{E}(\theta(w) - z_0)^2$ s.t. $\theta(w) \in \arg\min_\theta \sum_{k=1}^2 w_k \mathbb{E}(\theta - z_k)^2$,

where $z_0$, $z_1$ and $z_2$ follow normal distributions $\mathcal{N}(0,1)$, $\mathcal{N}(a,1)$ and $\mathcal{N}(-a,1)$, respectively, for any $a \neq 0$. Obviously, $w^* = (0.5, 0.5)$ is the optimal weight and $\theta(w^*) = 0 = \theta^*$. Throughout the paper, we assume $\theta^*$ can be learned by solving (P$_*$), which is stated formally below.

**Assumption 2** (Learnability of $\theta^*$ by (P$_*$)). *$\theta(w) = \theta^*$ for any $w \in \mathcal{W}^*$, where $\theta^*$ satsifes (P).*

Besides the situation like the aforementioned simple example, Assumption 2 holds obviously when $p_k = p_0$ for at least one $k \in \mathcal{K}$. In fact, the latter case happens when node 0 is a node in $\mathcal{K}$, so $w$ equal to one on that node and zero on others is optimal. Moreover, we will later on provide a refined generalization performance analysis for the latter case, so we state the latter case as a separate assumption below.

**Assumption 2′** (Existence of identical neighbors). *There exists a strict subset $\mathcal{J} \subset \mathcal{K}$ with $|\mathcal{J}| = J$ such that $p_k = p_0$ for $k \in \mathcal{J}$. Moreover,*

$$\mathcal{W}^* = \left\{ w \in \Delta_K^b \big| w_k = 0 \, for \, k \in \mathcal{K} \backslash \mathcal{J} \right\}. \tag{4}$$

The first statement in Assumption 2′ implies that the right-hand side of (4) is contained by the left-hand side. The second statement further assumes that they are equal. Assumption 2′ implies Assumption 2 because $\sum_{k=1}^{K} w_k L_k(\theta) = L_0(\theta)$ for any $\theta \in \Theta$ and any $w \in \mathcal{W}^*$ satisfying (4).

**Assumption 3** (Error bound condition). *There exist $C_r > 0$ and $r \geq 1$ such that*

$$\text{Dist}(w, \mathcal{W}^*) := \min_{w' \in \mathcal{W}^*} \|w - w'\| \leq C_r \left[ F(w) - \min_{w \in \Delta_K^b} F(w) \right]^{1/r}. \tag{5}$$

Inequality (5) means problem (P$_*$) satisfies the *error bound condition*, which has impact on the convergence property of many optimization algorithms (Johnstone & Moulin, 2020; Yang & Lin, 2018; Lewis & Pang, 1998; Pang, 1997; Lin et al., 2020). Due to the limit of space, we refer readers to Appendix B for a practical example satisfying Assumption 3.

We are interested in the generalization performance of $\widehat{\theta}(\widehat{w})$, represented by the gap $L_0(\widehat{\theta}(\widehat{w})) - L_0(\theta^*)$, as both $D^{\text{valid}}$ and $D^{\text{train}}$ grow. For simplicity of notation, we assume $n_k = n_{\text{train}}$ for any $k \in \mathcal{K}$ for some integer $n_{\text{train}} \gg n_{\text{valid}}$. To facilitate the analysis, we need to introduce a few notations. Given a probability measure $\mathbb{Q}$ on $\mathcal{Z}$, let $\mathcal{H} = \{l(\theta; \cdot) : \theta \in \Theta\}$ be a pseudometric metric space equipped with the pseudometric metric $\rho_{\mathbb{Q}}$, which is the $L_2$ distance metric with respect $\mathbb{Q}$, i.e., $\rho_{\mathbb{Q}}(l, l') := \sqrt{\int_{\mathcal{Z}} (l(z) - l'(z))^2 d\mathbb{Q}(z)}$ for $l, l' \in \mathcal{H}$. The ball with radius $\epsilon > 0$ centered at $l \in \mathcal{H}$ is defined as $B_{\epsilon}(l) := \{l' \in \mathcal{H} | \rho_{\mathbb{Q}}(l, l') \leq \epsilon\}$. Let $\mathcal{N}(\mathcal{H}; \rho_{\mathbb{Q}}, \epsilon)$ be the $\epsilon$-covering number of $\mathcal{H}$ with respect to $\rho_{\mathbb{Q}}$, i.e., $\mathcal{N}(\mathcal{H}; \rho_{\mathbb{Q}}, \epsilon) := \min\{m | \exists l_1, \ldots, l_m \in \mathcal{H}, \mathcal{H} \subset \cup_{i=1}^m B_{\epsilon}(l_i)\}$.

Following Chen et al. (2021a), we make the following assumption on $\mathcal{N}(\mathcal{H}; \rho_{\mathbb{Q}}, \epsilon)$, which is important for analyzing the generalization performance (Koltchinskii, 2006; Kakade et al., 2008).

**Assumption 4.** *There exist $C_{\mathcal{H}} > 0$ and $\nu_{\mathcal{H}} > 0$ such that, for any probability measure $\mathbb{Q}$ on $\mathcal{Z}$,*

$$\mathcal{N}(\mathcal{H}; \rho_{\mathbb{Q}}, \epsilon) \leq (C_{\mathcal{H}}/\epsilon)^{\nu_{\mathcal{H}}}, \quad \forall \epsilon > 0. \tag{6}$$

With the these assumptions, we obtain the following theorems whose proofs are in Appendix D.

**Theorem 1** (Bound independent of statistical distance). *Suppose Assumptions 1, 2 and 4 hold. There exists a universal constant[1] $C_g > 0$ such that, with a probability of at least $1 - \delta$,*

$$L_0(\widehat{\theta}(\widehat{w})) - L_0(\theta^*) \leq C_g \left( \frac{\nu_{\mathcal{H}} + \log(1/\delta)}{n_{valid}} \right)^{\frac{1}{2}} + C_g \frac{\ell_0}{\sqrt{\mu}} \left( \frac{\nu_{\mathcal{H}} + K + \log(1/\delta)}{n_{train}/(Kb^2)} \right)^{\frac{1}{4}}. \tag{7}$$

Under the same assumptions,[2] the generalization bound by Chen et al. (2021a) becomes

$$L_0(\widehat{\theta}(w)) - L_0(\theta^*) \leq C_g' \left( \frac{\nu_{\mathcal{H}} + \log(1/\delta)}{n_{\text{valid}}} \right)^{\frac{1}{2}} + C_g' \frac{\sqrt{\beta}}{\sqrt{\mu}} \left( \frac{\nu_{\mathcal{H}} + K \log(K) + \log(1/\delta)}{K n_{\text{train}}} \right)^{\frac{1}{4}}$$
$$+ L_0(\theta(w)) - L_0(\theta^*) \tag{8}$$

---

[1] We define a universal constant as a constant that does not depend on any parameter of the problem except $C_{\mathcal{H}}$ and $C_r$. This definition is made only to simply the constant factors in our bounds.

[2] A $(\rho, C_\rho)$-transferable assumption is needed in Chen et al. (2021a), which also holds in our case with $\rho = 2$ because of the Lipschitz continuity and strong convexity assumed in Assumption 1.

for a universal constant $C'_g$ and any $w$ satisfying $\beta^{-1} \leq w_k/w_j \leq \beta$ with $k \neq j$ for some $\beta > 0$. However, it is likely that the optimal solution $w^*$ has zero components (e.g., when $p_k = p_0$ for some $k$). If so, $\beta$ on the right-hand side of (8) needs to be arbitrarily large for $\widehat{w}(\approx w^*)$ in $(\widehat{P})$ to satisfy the aforementioned condition. Moreover, when $w = \widehat{w}$, the convergence of the last term $L_0(\theta(w)) - L_0(\theta^*)$ in (8) is not characterized in Chen et al. (2021a). On the contrary, Theorem 1 holds without any assumption on $\widehat{w}$ (zero components are allowed), does not depend on $\beta$ and provides a generalization bound converging in every term.[3] When $b = c/K$ with a constant $c \geq 1$, the right-hand side of (7) improves the first two terms on the right-hand side of (8) by a $\log(K)$ term.

The bounds (7) and (8) may both be dominated by the $O(1/\sqrt{n_{\text{valid}}})$ term, which is the same as the generalization bound achieved simply by learning locally with $D^{\text{valid}}$ (see Proposition 1 in Appendix E). However, if Assumption 3 holds and Assumption 2 is strengthened to Assumption 2', we can establish a generalization bound different from Theorem 1 which suggests that $(\widehat{P})$ can still outperform local training when the following statistical distance between $p_0$ and $p_k$'s is small:

$$G := \sqrt{\max_{\theta \in \Theta} \sum_{k \in \mathcal{K}} \left( L_0(\theta) - L_k(\theta) \right)^2}. \tag{9}$$

**Theorem 2** (Bound dependent on statistical distance). *Suppose Assumptions 1, 2', 3 and 4 hold. There exists universal constants $C_e > 0$ and $C_w > 0$ such that, with a probability of at least $1 - 3\delta$,*

$$\text{Dist}(\widehat{w}, \mathcal{W}^*) \leq \varepsilon(n_{valid}, n_{train}) := C_w \left( \frac{\nu_{\mathcal{H}} + \log(1/\delta)}{n_{valid}} \right)^{\frac{1}{2r}} + C_w \frac{\ell_0}{\sqrt{\mu}} \left( \frac{\nu_{\mathcal{H}} + K + \log(1/\delta)}{n_{train}/(Kb^2)} \right)^{\frac{1}{4r}} \tag{10}$$

*and*

$$L_0(\widehat{\theta}(\widehat{w})) - L_0(\theta^*) \leq C_e \sqrt{\frac{\nu_{\mathcal{H}} + J + \log(1/\delta)}{N_\varepsilon}} + C_e \frac{\varepsilon(n_{valid}, n_{train})(K - J)}{b\sqrt{N_\varepsilon}} + 2\varepsilon(n_{valid}, n_{train})G, \tag{11}$$

*where $N_\varepsilon = \frac{n_{train}}{b^2 J + \varepsilon^2(n_{valid}, n_{train})(K - J)}$ and $G$ is defined in (9).*

Note that $N_\varepsilon = \Theta(n_{\text{train}})$. Based on the decreasing rate of $\varepsilon(n_{\text{valid}}, n_{\text{train}})$ in (10), we simplify (11) by only showing the bounds in terms of $n_{\text{valid}}, n_{\text{train}}$ and $G$ for a clear comparison with local training.

**Corollary 1.** *Suppose the assumptions of Theorem 2 hold and $n_{valid}$ and $n_{train}$ are large enough such that $\varepsilon(n_{valid}, n_{train})$ defined in (10) satisfies $\varepsilon(n_{valid}, n_{train}) \leq \frac{b\sqrt{J}}{K-J}$. With a probability of at least $1 - 3\delta$, we have $L_0(\widehat{\theta}(\widehat{w})) - L_0(\theta^*) \leq O\left( 1/n_{train}^{\frac{1}{2}} + G \cdot \left( 1/n_{valid}^{\frac{1}{2r}} + 1/n_{train}^{\frac{1}{4r}} \right) \right).$*

When $G = o(1/n_{\text{valid}}^{\frac{1}{2} - \frac{1}{2r}})$ and $G = o(n_{\text{train}}^{\frac{1}{4r}}/n_{\text{valid}}^{\frac{1}{2}})$, the bound in Corollary 1 becomes $o(1/n_{\text{valid}}^{\frac{1}{2}})$, meaning that method $(\widehat{P})$ has a better generalization guarantee than training locally. Since $G$ is small in this case, a natural question is whether optimizing the weight in $(\widehat{P})$ is still needed because the FL with equally weighted nodes may already have a good performance with respect to $p_0$. However, we show in Proposition 2 in Appendix E that $(\widehat{P})$ is still preferred to FL with equally weighted nodes for any $G$. We show the impacts of $b$, $J$ and $K$ through Corollary 2 in Appendix D.

## 4 FEDERATED BILEVEL OPTIMIZATION ALGORITHM

Although our main focus is the generalization performance of $(\widehat{P})$, we present a federated optimization algorithm for $(\widehat{P})$ based on the existing techniques by Gorbunov et al. (2021) and Ghadimi & Wang (2018). Different from a single-level optimization problem, the outer objective $\widehat{F}(w)$ in $(\widehat{P})$ depends implicitly on $w$ through the inner optimal solution $\widehat{\theta}(w)$, which makes the exact gradient $\nabla \widehat{F}(w)$ difficult to compute. A commonly used solution is to exploit implicit function as shown in the following lemma, which is from Lemma 2.1 and 2.2 in Ghadimi & Wang (2018).

**Lemma 1.** *Under Assumption 1, $\nabla \widehat{F}(w)$ is $\ell_F$-Lipschitz continuous with*

$$\ell_F := \left( \frac{2\ell_0 \ell_1}{\mu} + \frac{\ell_2 \ell_0^2}{\mu^2} \right) \frac{\sqrt{K}\ell_0}{\mu} + \frac{K\ell_1 \ell_0^2}{\mu^2}, \tag{12}$$

---

[3]As a by-product of our analysis, we show in (38) that $L_0(\theta(\widehat{w})) - L_0(\theta^*)$ also satisfies (7).

---

**Algorithm 1:** Local-SVRG method for (16): `Local-SVRG`$(\{f_{k,i}\}, w, x^{(0)}, \gamma, \tau, q, T)$

---

1   **Input**: functions $\{f_{k,i}\}$, weight $w$, initial vector $x^{(0)} \in \mathbb{R}^d$, learning rate $\gamma$, communication period $\tau \geq 1$, probability $q$ of updating reference point, and the total number of iterations $T$

2   $x_k^{(0)} = x^{(0)}, y_k^{(0)} = x^{(0)}, k = 1, \ldots, K$

3   **for** $t = 0, 1, \ldots, T - 1$ **do**

4      **for** $k = 1, \ldots, K$ in parallel **do**

5          Choose $i_k$ from $\{1, \ldots, n_k\}$ uniformly at random

6          $g_k^{(t)} = \nabla f_{k,i_k}(x_k^{(t)}) - \nabla f_{k,i_k}(y_k^{(t)}) + \nabla f_k(y_k^{(t)})$

7          $y_k^{(t+1)} = x_k^{(t)}$ with probability $q$ and $y_k^{(t+1)} = y_k^{(t)}$ with probability $1 - q$

8          **if** $t + 1 \bmod \tau = 0$ **then**

9              $x_k^{(t+1)} = x^{(t+1)} := \sum_{k=1}^K w_k \left( x_k^{(t)} - \gamma g_k^{(t)} \right)$

10         **else**

11             $x_k^{(t+1)} = x_k^{(t)} - \gamma g_k^{(t)}$

12         **end**

13      **end**

14 **end**

15 **Return:** $\bar{x}^{(T)} = U_T^{-1} \sum_{t=0}^T u_t x^{(t)}$ with $u_t = (1 - \min\{\gamma\mu, q/4\})^{-(t+1)}$ and $U_T = \sum_{t=0}^T u_t$.

---

*and* $\nabla \widehat{F}(w) = (\nabla_k \widehat{F}(w))_{k=1,\ldots,K}$, *where* $\nabla_k \widehat{F}(w)$ *is the partial derivative of* $\widehat{F}$ *w.r.t.* $w_k$ *and*

$$\nabla_k \widehat{F}(w) = -\nabla \widehat{L}_k(\widehat{\theta}(w))^\top \left( \sum_{k=1}^K w_k \nabla^2 \widehat{L}_k(\widehat{\theta}(w)) \right)^{-1} \nabla \widehat{L}_0(\widehat{\theta}(w)). \tag{13}$$

By Lemma 1, computing $\nabla_k \widehat{F}(w)$ requires solving (1) exactly and taking the inverse of the Hessian matrix in (13), both of which are challenging. Hence, for a given $w$, we will find an approximate solution of (1), denoted by $\bar{\theta}(w)(\approx \widehat{\theta}(w))$, and approximate the matrix inversion in (13) by solving a strongly convex quadratic program. In particular, we will approximate $\nabla \widehat{F}(w)$ by

$$\bar{\nabla} \widehat{F}(w) := (\bar{\nabla}_k \widehat{F}(w))_{k=1,\ldots,K} \quad \text{with} \quad \bar{\nabla}_k \widehat{F}(w) := -\nabla \widehat{L}_k(\bar{\theta}(w))^\top \bar{h}, \tag{14}$$

$$\text{where} \qquad \bar{h} \approx \arg\min_h \tfrac{1}{2} h^\top \left( \sum_{k=1}^K w_k \nabla^2 \widehat{L}_k(\bar{\theta}(w)) \right) h - h^\top \nabla \widehat{L}_0(\bar{\theta}(w)). \tag{15}$$

Both (1) and (15) can be written as a *distributed finite-sum* minimization on $K$ weighted nodes:

$$\min_{x \in \mathbb{R}^d} f(x) := \sum_{k=1}^K w_k f_k(x), \quad \text{where} \quad f_k(x) = \tfrac{1}{n_k} \sum_{i=1}^{n_k} f_{k,i}(x), \quad k = 1, \ldots, K. \tag{16}$$

When $f_{k,i}(\theta) = l(\theta; z_k^{(i)})$, (16) becomes (1). When $f_{k,i}(h) = \tfrac{1}{2} h^\top \nabla^2 l(\bar{\theta}(w); z_k^{(i)}) h - h^\top \nabla \widehat{L}_0(\bar{\theta}(w))$, (16) becomes (15).

With this observation, we apply Local-SVRG by Gorbunov et al. (2021) to the aforementioned two instances (16) to obtain $\bar{\theta}(w)$ and $\bar{h}$, which are used to construct the approximate gradient $\bar{\nabla}\widehat{F}(w)$ in (14). Then we update $w$ using $\bar{\nabla}\widehat{F}(w)$ based on the accelerated bilevel approximation method (ABA) by Ghadimi & Wang (2018). We choose the combination of Local-SVRG and the ABA methods because it leads to the lowest communication complexity in literature for solving $(\widehat{P})$. We formally present this approach in Algorithms 1 and 2. Recall that we have assumed $D^{\text{valid}}$ is stored in node 0, which is called center in Algorithm 2.

In each iteration of Algorithm 2, in addition to the communication within Local-SVRG, constantly many rounds of communication are needed to exchange $\theta^{(s)}$, $h^{(s)}$ $\nabla \widehat{L}_0(\theta^{(s)})$ and $\nabla \widehat{L}_k(\theta^{(s)})$ between the center and node $k$. We present the communication complexity of Algorithm 2 which can be proved by adapting the analysis in Gorbunov et al. (2021) and Ghadimi & Wang (2018) to our setting. The proofs are deferred to Sections F.1 and F.2.

**Theorem 3.** *Suppose Assumption 1 holds and* $\widehat{F}(w)$. *Let* $R := \max_{w \in \Delta_K^b} \|\widehat{\theta}(w)\|$ *and*

$$\gamma_0 := \min \left\{ \frac{3}{80\ell_1}, \frac{1}{\ell_1 \sqrt{5e(\tau - 1)[6(\tau - 1) + 8 + 16/(1 - q)]}}, \frac{q}{4\mu} \right\}. \tag{17}$$

---

**Algorithm 2:** Federated Learning Method for Bilevel Optimization $(\widehat{P})$

---

1 **Input**: initial weight $w^{(0)}$, learning rate $\eta$, training data $D_k^{\text{train}}$ for $k \in \mathcal{K}$, validation data $D^{\text{valid}}$, the number of outer iterations $S$, parameters $(\gamma, \tau, q)$ for `Local-SVRG`, and the number of inner iterations $T_s$ for $s = 0, \ldots, S-1$

2 Set $w_{\text{ag}}^{(0)} = w^{(0)}$

3 **for** $s = 0, 1, \ldots, S-1$ **do**

4     $w_{\text{md}}^{(s)} = \frac{2}{s+2} w^{(s)} + \frac{s}{s+2} w_{\text{ag}}^{(s)}$

5     Compute $\bar{\nabla} \widehat{F}(w_{\text{md}}^{(s)})$ as follows:

6        Set $f_{k,i}(\theta) = l(\theta; z_k^{(i)}), \quad i = 1, \ldots, n_k, \quad k = 1, \ldots, K$

7        Compute $\theta^{(s)} = \texttt{Local-SVRG}(\{f_{k,i}\}, w_{\text{md}}^{(s)}, \theta^{(s-1)}, \gamma, \tau, q, T_s)$ and send it to each node.

8        Compute $\nabla \widehat{L}_0(\theta^{(s)})$ at center and send it to each node.

9        Set $f_{k,i}(h) = \frac{1}{2} h^\top \nabla^2 l(\theta^{(s)}; z_k^{(i)}) h - h^\top \nabla \widehat{L}_0(\theta^{(s)}), \quad i = 1, \ldots, n_k, \quad k = 1, \ldots, K$

10        Compute $h^{(s)} = \texttt{Local-SVRG}(\{f_{k,i}\}, w_{\text{md}}^{(s)}, \nabla \widehat{L}_0(\theta^{(s)}), \gamma, \tau, q, T_s)$ and send it to each node.

11        Each node computes $\nabla \widehat{L}_k(\theta^{(s)})$ in parallel and send it to the center.

12        Set $\bar{\nabla}_k \widehat{F}(w_{\text{md}}^{(s)}) = -\nabla \widehat{L}_k(\theta^{(s)})^\top h^{(s)}$ for $k = 1, \ldots, K$

13     $w^{(s+1)} = \arg\min_{w \in \Delta_K^b} \left\langle \bar{\nabla} \widehat{F}(w_{\text{md}}^{(s)}), w \right\rangle + \frac{2}{\eta(s+1)} \| w - w^{(s)} \|^2$

14     $w_{\text{ag}}^{(s+1)} = \arg\min_{w \in \Delta_K^b} \left\langle \bar{\nabla} \widehat{F}(w_{\text{md}}^{(s)}), w \right\rangle + \frac{1}{2\eta} \| w - w_{\text{md}}^{(s)} \|^2$

15 **end**

16 **Return:** $w_{\text{ag}}^{(S)}$

---

*Suppose $\eta = \frac{1}{3\ell_F}$ in Algorithm 2 with $\ell_F$ defined as in (12). There exist constants $A_1$, $A_2$ and $A_3$ that only depend on $\ell_0$, $\ell_1$, $\ell_2$, $\mu$, $R$, $q$ and $K$ but not on $\tau$ such that the following statements hold.*

- *Suppose $\tau = 1$, $\gamma = \gamma_0$ and $T_s = \frac{1}{\gamma_0 \mu} \ln\left(\frac{A_1(s+1)^4}{\gamma_0}\right)$. Algorithm 2 finds an $\epsilon$-optimal solution of $(\widehat{P})$ with $\tilde{O}\left(\epsilon^{-0.5}\right)$ rounds of communication.*

- *Suppose $\tau > 1$, $\gamma = \frac{1}{M_s}$ and $T_s = \mu^{-1} M_s \ln\left(M_s^3\right)$, where*

$$M_s = \max\left\{ 1/\gamma_0, (s+1)^2 \sqrt{[A_1 + A_2(\tau-1) + A_3(\tau-1)^2]} \right\}, s = 0, 1, \ldots. \tag{18}$$

*Algorithm 2 finds an $\epsilon$-optimal solution of $(\widehat{P})$ with $\tilde{O}\left(\epsilon^{-1.5}\right)$ rounds of communication.*

When $\widehat{F}$ in $(\widehat{P})$ is non-convex, we aim at finding an $\epsilon$-stationary point of $(\widehat{P})$. Following Ghadimi & Wang (2018), we apply a standard proximal gradient method to $(\widehat{P})$ based on the approximate gradient $\bar{\nabla} \widehat{F}(w)$ in (14). This method and its analysis are standard and we include them in Section F.3 due to the limit of space. In Remark 1 in Section F.3, we also show that the complexity of our method is lower than those of Li et al. (2022) and Tarzanagh et al. (2022).

## 5 NUMERICAL EXPERIMENT

In this section, we demonstrate the performance of our methods on image classification tasks. We compare our method, denoted by **Bi-level**, against four baselines, including (1) **Local-train**, which solves $\min_{\theta \in \Theta} \widehat{L}_0(\theta)$ locally; (2) **FedAvg** (McMahan et al., 2017), which solves (1) with $w_k = 1/K$; (3) **Ditto** (Li et al., 2021); and (4) **pFedMe** (T Dinh et al., 2020). Ditto and pFedMe are two personalized FL methods. We apply all methods to train a convolutional neural network (CNN) on multiple image datasets: Fashion-MNIST (Xiao et al., 2017), MNIST (Deng, 2012), CIFAR-10 (Krizhevsky et al., 2009) and downsampled $32 \times 32$ ImageNet (Chrabaszcz et al., 2017). We

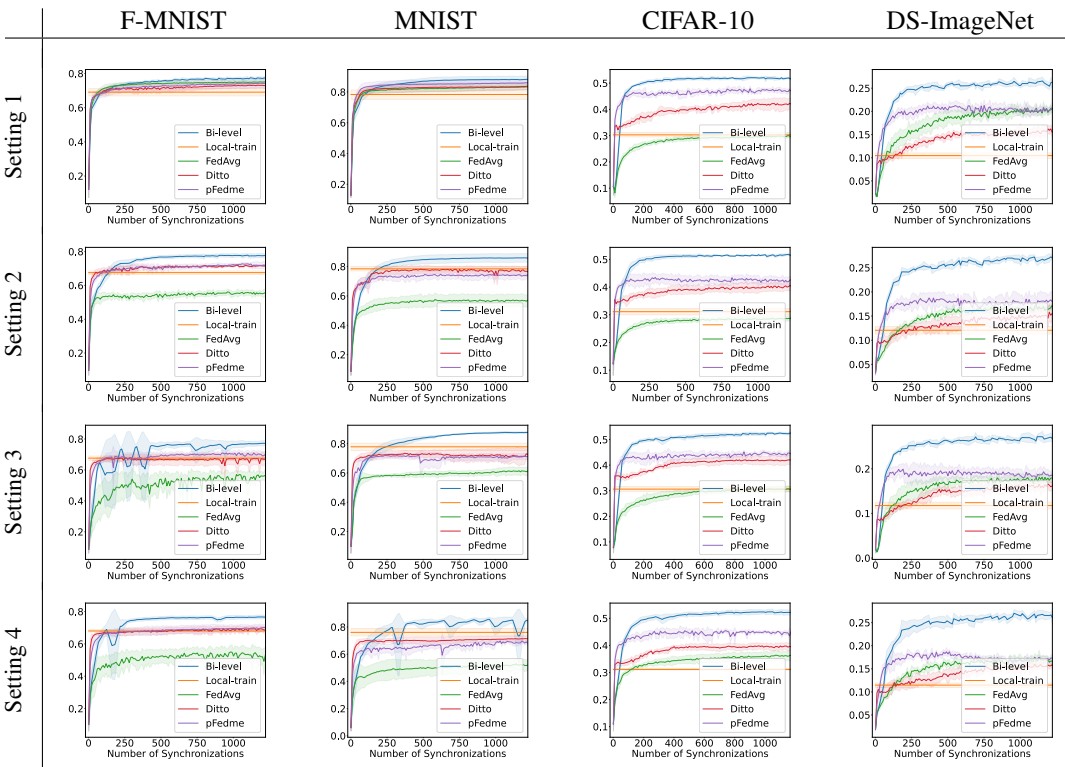

Figure 1: Comparison in test accuracy for the minority group vs number of synchronizations.

denote Fashion-MNIST and downsampled ImageNet by F-MNist and DS-ImageNet, respectively. See Appendix G for more details on the CNN and the computing environment we use.

We use a mini batch of size 50 to construct the stochastic gradients in all methods. In the Bi-level method, Algorithm 3 is applied to ($\widehat{\text{P}}$) with $b = 1/3$ and five epochs are performed within each call of Local-SVRG (i.e., $T_s = 5n_{\text{train}}/50$). We set $q = 1/50$, choose $\gamma$ from $\{0.05, 0.02, 0.01\}$ when solving (1) and from $\{0.0005, 0.0002, 0.0001\}$ when solving (15), choose $\tau$ from $\{10, 20\}$, and $\eta$ from $\{0.025, 0.02, 0.015\}$. We choose the combination that produces the highest validation accuracy after five outer iterations. SVRG is applied to $\min_{\theta \in \Theta} \widehat{L}_0(\theta)$ in Local-train and Local-SVRG is applied to (1) with $w_k = 1/K$ in FedAvg. Parameters $\tau$, $q$, and $\eta$ in FedAvg and Local-train are set the same as in our method. Ditto is implemented by setting $S_t = \mathcal{K}$, $r = \tau$, $s = 25$ and $\eta_g = \eta_l = \gamma$ in Algorithm 2 in Li et al. (2021), where $\tau$ and $\gamma$ are set the same as in our method. Similar to Li et al. (2021), we choose $\lambda$ in Ditto from $\{0.05, 0.1, 0.2\}$ to maximize the validation accuracy after five outer iterations. pFedMe is implemented by setting $\beta = 1$, $\delta = 0.005$, $R = \tau$ and $\eta = \gamma$ in Algorithm 1 in T Dinh et al. (2020) with $\tau$ and $\gamma$ set the same as in our method. Each subproblem in pFedMe is solved by gradient descend with a maximum iterations of 20. Like Li et al. (2021), we choose $\lambda$ in pFedMe from $\{5, 10, 15\}$ to maximize the validation accuracy after five outer iterations.

We set $\mathcal{K} = \{1, \ldots, 15\}$ (i.e., $K = 15$) and partition it into two groups, a minority group $\mathcal{J}_m = \{1, \ldots, 5\}$ and a majority group $\mathcal{J}_M = \{6, \ldots, 15\}$. We then generate $D_k^{\text{train}}$ for $k \in \mathcal{K}$ by randomly sampling data from the training sets with some artificial distributions, such that the data distributions (i.e., $p_k$'s) are the same within each group but different between groups. In particular, we create the data distributions of $\mathcal{J}_m$ and $\mathcal{J}_M$ under four different settings. In **Setting 1**, we create two different distributions over the classes and use them to sample $D_k^{\text{train}}$ with $k \in \mathcal{J}_m$ and $k \in \mathcal{J}_M$, respectively. In **Setting 2**, **Setting 3** and **Setting 4**, we first sample data in the same way as Setting 1 and, additionally, we permute the class labels among a few classes in $D_k^{\text{train}}$ with $k \in \mathcal{J}_M$ under Setting 2, rotate each image in $D_k^{\text{train}}$ with $k \in \mathcal{J}_M$ by 90 degrees in the same but random direction under Setting 3, and do both under Setting 4. This creates nodes with different levels of heterogeneity.

To compare the performances of the methods on both groups, we conduct two sets of experiments under each setting, one with $p_0$ being the distribution of $\mathcal{J}_m$ (i.e., $\mathcal{J} = \mathcal{J}_m$) and the other with

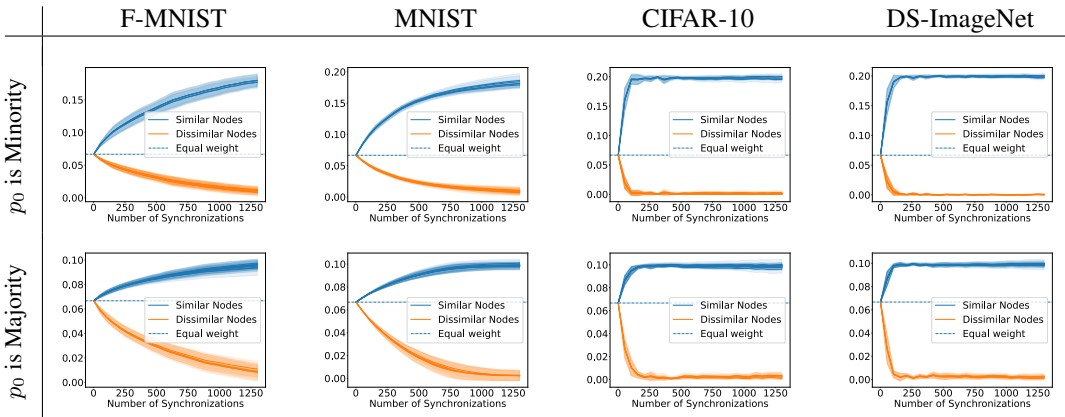

Figure 2: How $w$ evolves during the Bi-level method under Setting 1.

$p_0$ being the distribution of $\mathcal{J}_M$ (i.e., $\mathcal{J} = \mathcal{J}_M$). $D^{\text{valid}}$ is then sampled from $p_0$. For out-of-sample evaluation, we generate testing data by sampling from the testing set of each dataset using distribution $p_0$ described above under each setting. We denote the testing set by $D^{\text{test}}$ and let $n_{\text{test}} = |D^{\text{test}}|$. We repeat all experiments five times using different random seeds. The values of $n_{\text{valid}}$, $n_{\text{train}}$, $n_{\text{test}}$ and the details of data generation are presented in Sections G.1, G.2 and G.3.

We plot the test (top-1) accuracy each method obtains during iterations for the minority group in Figure 1, where the horizontal axis represents the number of synchronizations, i.e., the rounds of communications the method performs. Since Local-train does not require any communication, we just plot a horizontal line positioned at its final accuracy. Due to space limit, we present the accuracy for the majority group in Figure 7 in Section G.4. We also report the same results in Figure 8 and Figure 9 but the horizontal axis there represents the cumulative number of data points each method processes in parallel. In each figure, we show the confidence intervals of the curves as shaded areas.

According to Figure 1, our Bi-level method performs better than the four benchmarks on the minority group on all datasets under all settings. Local-train does not perform well because it only gets access to a small amount of data. The poor performance of FedAvg is because of the heterogeneity we created across nodes. In fact, FedAvg is even worse than Local-train in many cases, especially in Settings 2, 3 and 4 where the heterogeneity is high. This is consistent with the findings in literature. Although Ditto and pFedMe are designed for heterogeneous nodes, they still use a fixed weight on each node to train a global model, which may not provide a good starting point for personalization due to the high heterogeneity. In fact, their performances drop more or less as the data heterogeneity increases from Setting 1 to Settings 2, 3 and 4. On the contrary, by updating the weights, our method filters the information in the network and help the node in the minority group to find its similar peers and produce a good model through intra-group collaboration. Comparing Figure 1 with Figure 7, we find that the performances of FedAvg, Ditto and pFedMe are improved on the majority group. This is again because they utilize the information aggregated from all nodes, which is in favor of the majority. However, our method perform similarly on both groups and is still overall the best for the majority group. Similar phenomena are found in Figure 8 and Figure 9.

In addition, we also plot in Figure 2 how the weight $w_k$ for each node evolves during the Bi-level method under Setting 1. We show the results when $p_0$ is the distribution of the majority and the minority groups separately. In each case, we call the nodes in $\mathcal{J}$ similar nodes (to node 0) and call the others dissimilar nodes. According to Figure 2, our method successfully detects similar nodes in both cases and increases their weights but decreases the weights of dissimilar nodes. We present the weights under Settings 2, 3 and 4 in Section G.4. Similar phenomenons are observed.

## 6 CONCLUSION

We propose a FL approach on a network with weighted nodes and develop a federated bilevel optimization algorithm to optimize the weights based on the model's performance on a validation set. We analyze the generalization performance of the resulting model and identify the scenarios where our method theoretically outperforms training with local data and FL with even weights.

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

## A  SUMMARY OF NOTATIONS

| Symbol | Definition |
|---|---|
| $\mathcal{Z}$ | Sample space |
| $p_0$ | An unknown ground truth distribution on $\mathcal{Z}$ |
| $D^{\text{valid}}$ | Validation set i.i.d. sampled from $p_0$ and stored in node $0$ (or called *center*) |
| $n_0$ | Size of validation set also denoted by $n_{\text{valid}}$ |
| $\mathcal{K}$ | Set of nodes defined by $\{1, \ldots, K\}$ |
| $\mathcal{J}$ | A strict subset of $\mathcal{K}$ |
| $J$ | Size of $\mathcal{J}$ |
| $p_k$ | The distribution on $\mathcal{Z}$ in node $k$, $k = 1, \ldots, K$ |
| $D_k^{\text{train}}$ | Training set i.i.d. sampled from $p_k$ and stored in node $k$, $k = 1, \ldots, K$ |
| $n_k$ | Size of training set $D_k^{\text{train}}$ |
| $w$ | Vector of weights defined by $(w_1, \ldots, w_K) \in [0,1]^K$ |
| $b$ | A user-defined parameter in $[\frac{1}{K}, 1]$ |
| $\Delta_K^b$ | Capped simplex where $w$ comes from |
| $\theta$ | Vector of model parameters |
| $\Theta$ | Set where $\theta$ comes from |
| $l(\theta; z)$ | Loss of $\theta$ on a data point $z$ from the space $\mathcal{Z}$ |
| $\widehat{L}_0(\theta)$ | Empirical loss of $\theta$ on $D^{\text{valid}}$ |
| $\widehat{L}_k(\theta)$ | Empirical loss of $\theta$ on $D_k^{\text{train}}$, $k = 1, \ldots, K$ |
| $L_0(\theta)$ | Generalization loss of $\theta$ on $p_0$ |
| $L_k(\theta)$ | Generalization loss of $\theta$ on $p_k$, $k = 1, \ldots, K$ |
| $\ell_1$ | Lipschitz constant of $l(\theta; z) \in [0,1]$ and $\nabla l(\theta; z)$ in $\theta$ for any $z \in \mathcal{Z}$ |
| $\ell_0$ | Lipschitz constant of all $\widehat{L}_k(\theta)$ for $k \in \mathcal{K}$ |
| $\ell_2$ | Lipschitz constant of all $\nabla^2 \widehat{L}_k(\theta)$ for $k \in \mathcal{K}$ |
| $\mu$ | parameter of strong convexity of $\widehat{L}_k(\theta)$ for $k \in \mathcal{K}$ |
| $G$ | Statistical distance between $p_0$ and $p_k$'s defined in (9) |
| $\widehat{F}(w)$ | The objective in the bilevel optimization problem $(\widehat{P})$ |
| $F(w)$ | The objective in the auxiliary problem $(P_*)$ |
| $\widehat{\theta}(w)$ | Optimal solution of FL on weighted nodes formulated in (1) |
| $\theta^*$ | Optimal solution of the learning problem $(P)$ |
| $\mathcal{W}^*$ | Optimal solution set of the auxiliary problem $(P_*)$ |
| $\widehat{w}$ | Optimal solution of the bilevel optimization problem $(\widehat{P})$ |
| $\gamma$ | Learning rate |
| $\tau$ | Communication period |
| $q$ | Probability of updating reference point in the Local-SVRG method |
| $S$ | Number of stages |
| $T_s$ | Number of iterations for stage $s \in \{0, 1, \ldots, S-1\}$ |

Table 1: Notations used throughout the paper.

## B  EXAMPLES SATISFYING ASSUMPTION 3

We consider $(P_*)$ in the setting of linear regression. Consider data $z = (x, y)$, where $x \in \mathbb{R}^d$ is a feature vector and $y \in \mathbb{R}$ is a continuous target variable, and consider the quadratic loss $l(\theta; z) = \frac{1}{2}(x^\top \theta - y)^2$. We assume $x$ in all nodes, including node $0$ (center) and the nodes in $\mathcal{K}$, follows the same distribution, and matrix $\mathbb{E}[xx^\top]$ is non-singular. Moreover, we assume that there is a vector $\theta_k^* \in \mathbb{R}^d$ associated to node $k$, and $y$ in node $k$ is generated as $y = x^\top \theta_k^* + \epsilon_k$ for $k = 0, 1, \ldots, K$, where $\epsilon_k$ is a zero-mean random noise indepedent of $x$. In this problem, we have

$$L_k(\theta) = \tfrac{1}{2}\mathbb{E}\big[(x^\top \theta - y)^2\big] = \tfrac{1}{2}\mathbb{E}\big[(x^\top \theta - x^\top \theta_k^* - \epsilon_k)^2\big] = \tfrac{1}{2}(\theta - \theta_k^*)^\top \mathbb{E}\big[xx^\top\big](\theta - \theta_k^*) + \tfrac{1}{2}\mathbb{E}\big[\epsilon_k^2\big].$$

We can easily show that $\theta(w)$ in $(P_*)$ has the closed form $\theta(w) = \sum_{k=1}^K w_k \theta_k^*$, which means

$$F(w) = L_0(\theta(w)) = \tfrac{1}{2}(\sum_{k=1}^K w_k \theta_k^* - \theta_0^*)^\top \mathbb{E}\big[xx^\top\big](\sum_{k=1}^K w_k \theta_k^* - \theta_0^*) + \tfrac{1}{2}\mathbb{E}\big[\epsilon_0^2\big].$$

This is a quadratic function of $w$ over the polyhedral set $\Delta_K^b$ and thus satisfies the error bound condition (5) with $r = 2$ according to Lemma 1 in Gong & Ye (2014).

## C  Technical Lemmas

In this section, we provide some technical lemmas with proofs which are necessary for establishing the main theorems. The main steps in the proofs of Lemma 2 and 3 are borrowed from Chen et al. (2021a). However, the generalization bound in Theorem 3.1 in Chen et al. (2021a) is for a $w$ satisfying $\beta^{-1} \leq w_k/w_j \leq \beta$ with $k \neq j$ for some $\beta > 0$, and their bound increases with $\beta$. When applied to $w = \widehat{w}$ with zero or nearly zero components (that happens when $p_k = p_0$ for some $k$), such a $\beta$ is very large or equals infinity. Therefore, we make necessary changes in the proofs to extend the results for a generic $w$ and a $w$ that is $\varepsilon$-away from $\mathcal{W}^*$ (see Lemma 3) where the components can be nearly zero. These extensions are important for proofing our main theorems.

**Lemma 2.** *Suppose Assumptions 1 and 4 hold. There exists a universal constant $C_0 > 0$ such that, with a probability of at least $1 - \delta$,*

$$\sup_{\theta \in \Theta} \left| L_0(\theta) - \widehat{L}_0(\theta) \right| \leq C_0 \sqrt{\frac{\nu_{\mathcal{H}} + \log(1/\delta)}{n_{valid}}}.$$

*Proof.* For simplicity of notation, we write $n_{\text{valid}}$ as $n$ in this proof. Let

$$G_{\text{valid}}(D^{\text{valid}}) := \sup_{\theta \in \Theta} \left[ L_0(\theta) - \widehat{L}_0(\theta) \right] \quad \text{and} \quad G'_{\text{valid}}(D^{\text{valid}}) := \sup_{\theta \in \Theta} \left[ \widehat{L}_0(\theta) - L_0(\theta) \right].$$

Consider any $i \in \{1, 2, \ldots, n\}$. Let $D_i^{\text{valid}}$ be the same as $D^{\text{valid}}$ except that $z^{(i)}$ is replaced by another data point $z'^{(i)}$ sampled from $p_0$. Recall (3). We have

$$\left| G_{\text{valid}}(D^{\text{valid}}) - G_{\text{valid}}(D_i^{\text{valid}}) \right|$$

$$= \left| \sup_{\theta \in \Theta} \left[ L_0(\theta) - \widehat{L}_0(\theta) \right] - \sup_{\theta \in \Theta} \left[ L_0(\theta) - \widehat{L}_0(\theta) + \frac{1}{n} \left( l(\theta; z^{(i)}) - l(\theta; z'^{(i)}) \right) \right] \right| \leq \frac{1}{n},$$

where the inequality is because the loss is in $[0, 1]$ (Assumption 1). This inequality means we can apply the McDiarmid's inequality to obtain that, for any $\epsilon > 0$,

$$\mathbb{P}\left( G_{\text{valid}}(D^{\text{valid}}) \geq \mathbb{E}[G_{\text{valid}}(D^{\text{valid}})] + \epsilon \right) \leq \exp(-2\epsilon^2 n),$$

or equivalently, with a probability of at least $1 - \delta$,

$$G_{\text{valid}}(D^{\text{valid}}) \leq \mathbb{E}[G_{\text{valid}}(D^{\text{valid}})] + \sqrt{\frac{\log(1/\delta)}{2n}}.$$

Next, we apply the standard symmetrization argument by introducing a ghost dataset

$$D_{\text{ghost}}^{\text{valid}} := \left\{ z'^{(i)} \right\}_{i=1}^n,$$

which is independent of $D^{\text{valid}}$ and sampled from $p_0$. Let $\{\sigma_i\}_{i=1}^n$ be Rademacher random variables. We have

$$\mathbb{E}[G_{\text{valid}}(D^{\text{valid}})] = \mathbb{E}\left[ \sup_{\theta \in \Theta} \left[ L_0(\theta) - \widehat{L}_0(\theta) \right] \right] = \mathbb{E}\left[ \sup_{\theta \in \Theta} \left[ \mathbb{E}\left[ L(\theta; D_{\text{ghost}}^{\text{valid}}) \right] - L(\theta; D^{\text{valid}}) \right] \right]$$

$$\leq \mathbb{E}\left[ \sup_{\theta \in \Theta} \left[ L(\theta; D_{\text{ghost}}^{\text{valid}}) - L(\theta; D^{\text{valid}}) \right] \right] = \mathbb{E}\left[ \sup_{\theta \in \Theta} \frac{1}{n} \sum_{i=1}^n \sigma_i \left( l(\theta; z'^{(i)}) - l(\theta; z^{(i)}) \right) \right]$$

$$\leq 2\mathbb{E}\left[ \sup_{\theta \in \Theta} \frac{1}{n} \sum_{i=1}^n \sigma_i l(\theta; z^{(i)}) \right] = 2\mathbb{E}\widehat{R}_n(\mathcal{H}),$$

where $\widehat{R}_n(\mathcal{H}) = \mathbb{E}\left[ \sup_{\theta \in \Theta} \frac{1}{n} \sum_{i=1}^n \sigma_i l(\theta; z^{(i)}) \middle| D^{\text{valid}} \right]$ and $\mathcal{H} = \{l(\theta; \cdot) : \theta \in \Theta\}$.

Let $M_\theta := \frac{1}{\sqrt{n}} \sum_{i=1}^n \sigma_i l(\theta; z^{(i)})$ for $\theta \in \Theta$. By Hoeffding's Lemma, we have for any $\theta, \theta' \in \Theta$,

$$
\begin{aligned}
\mathbb{E}\left[\exp\left(\lambda(M_\theta - M_{\theta'})\right)\Big| D^{\text{valid}}\right] &= \prod_{i=1}^n \mathbb{E}\left[\exp\left(\frac{\lambda}{\sqrt{n}}\sigma_i\Big(l(\theta; z^{(i)}) - l(\theta'; z^{(i)})\Big)\right)\Big| D^{\text{valid}}\right] \\
&\le \prod_{i=1}^n \exp\left(\frac{\lambda^2}{2n}\Big(l(\theta; z^{(i)}) - l(\theta'; z^{(i)})\Big)^2\right) = \exp\left(\frac{\lambda^2}{2}\mathrm{d}^2(\theta, \theta')\right),
\end{aligned}
$$

where

$$
\mathrm{d}(\theta, \theta') = \sqrt{\sum_{i=1}^n \frac{1}{n}\Big(l(\theta; z^{(i)}) - l(\theta'; z^{(i)})\Big)^2} \le 1
$$

is the $L_2$-distance between mappings $l(\theta; \cdot)$ and $l(\theta'; \cdot)$ with respect to the empirical distribution over $D^{\text{valid}}$ and is a pseudometric in $\mathcal{H}$. Hence, by Dudley's entropy integral inequality (see Corollary 13.2 in Boucheron et al. (2013)), there exists a universal constant $C_d$ such that

$$
\widehat{R}_n(\mathcal{H}) = \frac{1}{\sqrt{n}}\mathbb{E}\left[\sup_{\theta \in \Theta} M_\theta \Big| D^{\text{valid}}\right] \le \frac{C_d}{\sqrt{n}}\int_0^1 \sqrt{\log\left(\mathcal{N}(\mathcal{H}; \mathrm{d}; \epsilon)\right)}d\epsilon
$$

According to Assumption 4, we have

$$
\mathbb{E}[G_{\text{train}}] \le 2\widehat{R}_n(\mathcal{H}) \le \frac{2C_d}{\sqrt{n}}\int_0^1 \sqrt{\nu_\mathcal{H}\log\left(\frac{C_\mathcal{H}}{\epsilon}\right)}d\epsilon
$$

Hence, with a probability of at least $1 - \delta$,

$$
G_{\text{valid}}(D^{\text{valid}}) \le \frac{2C_d}{\sqrt{n}}\int_0^1 \sqrt{\nu_\mathcal{H}\log\left(\frac{C_\mathcal{H}}{\epsilon}\right)}d\epsilon + \sqrt{\frac{\log(1/\delta)}{2n}}.
$$

Applying the same argument to $G'_{\text{valid}}(D^{\text{valid}})$, we can show that, with a probability of at least $1 - \delta$

$$
G'_{\text{valid}}(D^{\text{valid}}) \le \frac{2C_d}{\sqrt{n}}\int_0^1 \sqrt{\nu_\mathcal{H}\log\left(\frac{C_\mathcal{H}}{\epsilon}\right)}d\epsilon + \sqrt{\frac{\log(1/\delta)}{2n}}.
$$

By a union bound, we have, with a probability of at least $1 - \delta$

$$
\sup_{\theta \in \Theta}\left|L_0(\theta) - \widehat{L}_0(\theta)\right| \le \frac{2C_d}{\sqrt{n}}\int_0^1 \sqrt{\nu_\mathcal{H}\log\left(\frac{C_\mathcal{H}}{\epsilon}\right)}d\epsilon + \sqrt{\frac{\log(2/\delta)}{2n}},
$$

which completes the proof.

$\square$

Given $\varepsilon > 0$, we define

$$
\mathcal{W}_\varepsilon^* := \left\{w \in \Delta_K^b \Big| \text{Dist}(w, \mathcal{W}^*) \le \varepsilon\right\} \tag{19}
$$

$$
N_\varepsilon := \frac{n_{\text{train}}}{b^2 J + \varepsilon^2(K - J)}. \tag{20}
$$

**Lemma 3.** *Suppose Assumptions 1,2 and 4 hold. There exists a universal constant $C_a > 0$ such that, with a probability of at least $1 - \delta$,*

$$
\sup_{\theta \in \Theta, w \in \Delta_K^b}\left|\sum_{k=1}^K w_k\left[\widehat{L}_k(\theta) - L_k(\theta)\right]\right| \le C_a\sqrt{\frac{\nu_\mathcal{H} + K + \log(1/\delta)}{n_{train}/(Kb^2)}}. \tag{21}
$$

*Suppose Assumptions 1, 2', 3 and 4 hold. There exists a universal constant $C_a' > 0$ such that, with a probability of at least $1 - \delta$,*

$$
\sup_{\theta \in \Theta, w \in \mathcal{W}_\varepsilon^*}\left|\sum_{k=1}^K w_k\left[\widehat{L}_k(\theta) - L_k(\theta)\right]\right| \le C_a'\left(\sqrt{\frac{\nu_\mathcal{H} + J + \log(1/\delta)}{N_\varepsilon}} + \frac{\varepsilon(K - J)}{b\sqrt{N_\varepsilon}}\right). \tag{22}
$$

*Proof.* We prove (22) first. Suppose Assumptions 1, $2'$, 3 and 4 hold. By Assumptions $2'$, we have $w_j^* = 0$ for $w \in \mathcal{W}^*$ and $j \in \mathcal{K}\backslash\mathcal{J}$, which means $w_j \leq \sqrt{\sum_{k \in \mathcal{K}\backslash\mathcal{J}} w_k^2} \leq \mathrm{Dist}(w, \mathcal{W}^*) \leq \varepsilon$ for any $w \in \mathcal{W}_\varepsilon^*$ and $j \in \mathcal{K}\backslash\mathcal{J}$.

Let

$$G_{\mathrm{train}}(D^{\mathrm{train}}) := \sup_{\theta \in \Theta, w \in \mathcal{W}_\varepsilon^*} \left\{ \sum_{k=1}^{K} w_k \left[ L_k(\theta) - \widehat{L}_k(\theta) \right] \right\} \qquad (23)$$

$$G'_{\mathrm{train}}(D^{\mathrm{train}}) := \sup_{\theta \in \Theta, w \in \mathcal{W}_\varepsilon^*} \left\{ \sum_{k=1}^{K} w_k \left[ \widehat{L}_k(\theta) - L_k(\theta) \right] \right\}. \qquad (24)$$

Consider an index $j \in \{1, 2, \ldots, K\}$ and $i \in \{1, 2, \ldots, n_j\}$. Let $D_{j,i}^{\mathrm{train}}$ be the same as $D^{\mathrm{train}}$ except that $z_j^{(i)}$ is replaced by another data point $z_j'^{(i)}$ sampled from $p_j$. Recall (2). We have

$$\left| G_{\mathrm{train}}(D^{\mathrm{train}}) - G_{\mathrm{train}}(D_{j,i}^{\mathrm{train}}) \right|$$

$$= \left| \sup_{\theta \in \Theta, w \in \mathcal{W}_\varepsilon^*} \left\{ \sum_{k=1}^{K} w_k \left[ L_k(\theta) - \widehat{L}_k(\theta) \right] \right\} \right.$$

$$\left. - \sup_{\theta \in \Theta, w \in \mathcal{W}_\varepsilon^*} \left\{ \sum_{k=1}^{K} w_k \left[ L_k(\theta) - \widehat{L}_k(\theta) \right] + \frac{w_j}{n_j} \left( l\left(\theta; z_j^{(i)}\right) - l\left(\theta; z_j'^{(i)}\right) \right) \right\} \right|$$

$$\leq \frac{w_j}{n_j} \leq \begin{cases} \frac{b}{n_j} & j \in \mathcal{J} \\ \frac{\varepsilon}{n_j} & j \in \mathcal{K}\backslash\mathcal{J}. \end{cases} \qquad (25)$$

With this inequality, we can apply the McDiarmid's inequality to show that, for any $\epsilon > 0$,

$$\mathbb{P}\left( G_{\mathrm{train}}(D^{\mathrm{train}}) \geq \mathbb{E}[G_{\mathrm{train}}(D^{\mathrm{train}})] + \epsilon \right) \leq \exp\left( \frac{-2\epsilon^2 n_{\mathrm{train}}}{b^2 J + \varepsilon^2 (K - J)} \right),$$

which implies that, with a probability of at least $1 - \delta$,

$$G_{\mathrm{train}}(D^{\mathrm{train}}) \leq \mathbb{E}[G_{\mathrm{train}}(D^{\mathrm{train}})] + \sqrt{\frac{\log(1/\delta)}{N_\varepsilon}}. \qquad (26)$$

Next, we apply the standard symmetrization strategy by introducing a ghost dataset $D_{\mathrm{ghost}}^{\mathrm{train}} := \left\{ D_{k,\mathrm{ghost}}^{\mathrm{train}} \right\}_{k=1}^{K}$ where

$$D_{k,\mathrm{ghost}}^{\mathrm{train}} := \left\{ z_k'^{(i)} \right\}_{i=1}^{n_k} \quad \text{for } k = 1, \ldots, K$$

is a dataset independent of $D_k^{\mathrm{train}}$ sampled from $p_k$. Let $\{\sigma_{k,i}\}_{i=1}^{n_k}$ for $k = 1, \ldots, K$ be Rademacher random variables. We have

$$\mathbb{E}[G_{\mathrm{train}}(D^{\mathrm{train}})] = \mathbb{E}\left[ \sup_{\theta \in \Theta, w \in \mathcal{W}_\varepsilon^*} \left\{ \sum_{k=1}^{K} w_k \left[ L_k(\theta) - \widehat{L}_k(\theta) \right] \right\} \right]$$

$$= \mathbb{E}\left[ \sup_{\theta \in \Theta, w \in \mathcal{W}_\varepsilon^*} \left\{ \sum_{k=1}^{K} w_k \left( \mathbb{E}\left[ L_k(\theta; D_{k,\mathrm{ghost}}^{\mathrm{train}}) \right] - \widehat{L}_k(\theta) \right) \right\} \right]$$

$$\leq \mathbb{E}\left[ \sup_{\theta \in \Theta, w \in \mathcal{W}_\varepsilon^*} \left\{ \sum_{k=1}^{K} w_k \left( L_k(\theta; D_{k,\mathrm{ghost}}^{\mathrm{train}}) - \widehat{L}_k(\theta) \right) \right\} \right]$$

$$= \mathbb{E}\left[ \sup_{\theta \in \Theta, w \in \mathcal{W}_\varepsilon^*} \sum_{k=1}^{K} \sum_{i=1}^{n_k} \frac{\sigma_{k,i} w_k}{n_k} \left( l(\theta; z_k'^{(i)}) - l(\theta; z_k^{(i)}) \right) \right]$$

$$\leq 2\mathbb{E}\left[ \sup_{\theta \in \Theta, w \in \mathcal{W}_\varepsilon^*} \sum_{k=1}^{K} \sum_{i=1}^{n_k} \frac{\sigma_{k,i} w_k}{n_k} l(\theta; z_k^{(i)}) \right] = 2\mathbb{E}\widehat{R}_{n_{\mathrm{train}}}(\mathcal{H}, \mathcal{W}_\varepsilon^*), \quad (27)$$

where $\widehat{R}_{n_{\text{train}}}(\mathcal{H}, \mathcal{W}_\varepsilon^*) := \mathbb{E}\left[\sup_{\theta \in \Theta, w \in \mathcal{W}_\varepsilon^*} \sum_{k=1}^{K} \sum_{i=1}^{n_k} \frac{\sigma_{k,i} w_k}{n_k} l(\theta; z_k^{(i)}) \Big| D^{\text{train}}\right]$ and $\mathcal{H} = \left\{l(\theta; \cdot) : \theta \in \Theta\right\}$.

Let $M_{\theta,w} := \sqrt{N_\varepsilon} \sum_{k=1}^{K} \sum_{i=1}^{n_k} \frac{\sigma_{k,i} w_k}{n_k} l(\theta; z_k^{(i)})$ for $\theta \in \Theta$ and $w \in \mathcal{W}_\varepsilon^*$. Hence, by Hoeffding's Lemma, we have

$$
\begin{aligned}
&\mathbb{E}\left[\exp\left(\lambda(M_{\theta,w} - M_{\theta',w'})\right) \Big| D^{\text{train}}\right] \\
&= \prod_{k=1}^{K} \prod_{i=1}^{n_k} \mathbb{E}\left[\exp\left(\lambda \sqrt{N_\varepsilon} \frac{\sigma_{k,i}}{n_k}\left(w_k l(\theta; z_k^{(i)}) - w_k' l(\theta'; z_k^{(i)})\right)\right) \Big| D^{\text{train}}\right] \\
&\leq \prod_{k=1}^{K} \prod_{i=1}^{n_k} \exp\left(\frac{\lambda^2 N_\varepsilon}{2n_k^2}\left(w_k l(\theta; z_k^{(i)}) - w_k' l(\theta'; z_k^{(i)})\right)^2\right) \\
&= \exp\left(\frac{\lambda^2}{2} \mathrm{d}^2(\theta, w, \theta', w')\right),
\end{aligned}
\tag{28}
$$

where

$$
\mathrm{d}(\theta, w, \theta', w') = \sqrt{\sum_{k=1}^{K} \sum_{i=1}^{n_k} \frac{N_\varepsilon}{n_k^2}\left(w_k l(\theta; z_k^{(i)}) - w_k' l(\theta'; z_k^{(i)})\right)^2} \leq \sqrt{\sum_{k \in \mathcal{J}} \frac{N_\varepsilon b^2}{n_{\text{train}}} + \sum_{k \in \mathcal{K} \setminus \mathcal{J}} \frac{N_\varepsilon \varepsilon^2}{n_{\text{train}}}} = 1
$$

is a pseudo distance metric between $(l(\theta; \cdot), w)$ and $(l(\theta'; \cdot), w')$ in $\mathcal{H} \times \mathcal{W}_\varepsilon^*$. Hence, by Dudley's entropy integral inequality (see Corollary 13.2 in Boucheron et al. (2013)), there exists a universal constant $C_d$ such that

$$
\widehat{R}_{n_{\text{train}}}(\mathcal{H}, \mathcal{W}_\varepsilon^*) = \frac{1}{\sqrt{N_\varepsilon}} \mathbb{E}\left[\sup_{\theta \in \Theta, w \in \mathcal{W}_\varepsilon^*} M_{\theta,w} \Big| D^{\text{train}}\right] \leq \frac{C_d}{\sqrt{N_\varepsilon}} \int_0^1 \sqrt{\log\left(\mathcal{N}(\mathcal{H} \times \mathcal{W}_\varepsilon^*; \mathrm{d}; \epsilon)\right)} d\epsilon,
$$

where $\mathcal{N}(\mathcal{H} \times \mathcal{W}_\varepsilon^*; \mathrm{d}; \epsilon)$ is the $\epsilon$-covering number of $\mathcal{H} \times \mathcal{W}_\varepsilon^*$ w.r.t. $\mathrm{d}$.

We next need to bound $\mathcal{N}(\mathcal{H} \times \mathcal{W}_\varepsilon^*; \mathrm{d}; \epsilon)$. Note that

$$
\begin{aligned}
&\mathrm{d}^2(\theta, w, \theta', w') \\
&= \sum_{k=1}^{K} \sum_{i=1}^{n_k} \frac{N_\varepsilon}{n_k^2}\left(w_k l(\theta; z_k^{(i)}) - w_k' l(\theta'; z_k^{(i)})\right)^2 \\
&\leq \sum_{k=1}^{K} \sum_{i=1}^{n_k} \frac{2N_\varepsilon}{n_k^2} w_k^2\left(l(\theta; z_k^{(i)}) - l(\theta'; z_k^{(i)})\right)^2 + \sum_{k=1}^{K} \sum_{i=1}^{n_k} \frac{2N_\varepsilon}{n_k^2}(w_k - w_k')^2 l^2(\theta'; z_k^{(i)}) \\
&\leq \sum_{k \in \mathcal{J}} \sum_{i=1}^{n_k} \frac{2N_\varepsilon}{n_k^2} b^2\left(l(\theta; z_k^{(i)}) - l(\theta'; z_k^{(i)})\right)^2 + \sum_{k \in \mathcal{K} \setminus \mathcal{J}} \sum_{i=1}^{n_k} \frac{2N_\varepsilon}{n_k^2} \varepsilon^2\left(l(\theta; z_k^{(i)}) - l(\theta'; z_k^{(i)})\right)^2 \\
&\quad + \sum_{k=1}^{K} \sum_{i=1}^{n_k} \frac{2N_\varepsilon}{n_k^2}(w_k - w_k')^2 \\
&\leq \frac{2N_\varepsilon}{n_{\text{train}}}\left(\sum_{k \in \mathcal{J}} \sum_{i=1}^{n_{\text{train}}} \frac{b^2}{n_{\text{train}}}\left(l(\theta; z_k^{(i)}) - l(\theta'; z_k^{(i)})\right)^2 + \sum_{k \in \mathcal{K} \setminus \mathcal{J}} \sum_{i=1}^{n_{\text{train}}} \frac{\varepsilon^2}{n_{\text{train}}}\left(l(\theta; z_k^{(i)}) - l(\theta'; z_k^{(i)})\right)^2\right) \\
&\quad + \frac{2N_\varepsilon}{n_{\text{train}}}\|w - w'\|^2.
\end{aligned}
\tag{29}
$$

We then define a probability measure

$$
\mathbb{Q} = \frac{N_\varepsilon}{n_{\text{train}}}\left(\sum_{k \in \mathcal{J}} \sum_{i=1}^{n_{\text{train}}} \frac{b^2}{n_{\text{train}}} \delta_{z_k^{(i)}} + \sum_{k \in \mathcal{K} \setminus \mathcal{J}} \sum_{i=1}^{n_{\text{train}}} \frac{\varepsilon^2}{n_{\text{train}}} \delta_{z_k^{(i)}}\right)
\tag{30}
$$

on $\mathcal{Z}$, where $\delta_{z_k^{(i)}}$ is a point mass at $z_k^{(i)}$. Then, we can construct an $\epsilon$-cover for $\mathcal{H} \times \mathcal{W}_\varepsilon^*$ w.r.t. $\mathrm{d}$ by taking the Cartesian product of an $\frac{\epsilon}{2}$-cover for $\mathcal{H}$ w.r.t. distance metric $\rho_{\mathbb{Q}}(l, l') = \sqrt{\int_{\mathcal{Z}} (l(z) - l'(z))^2 d\mathbb{Q}(z)}$ for $l, l' \in \mathcal{H}$ and a $\sqrt{\frac{n_{\text{train}}}{N_\varepsilon}} \frac{\epsilon}{2}$-cover for $\mathcal{W}_\varepsilon^*$ w.r.t. the Euclidean distance. According to Assumption 4, the former has a cardinality of $(2C_{\mathcal{H}}/\epsilon)^{\nu_{\mathcal{H}}}$. To construct the latter, we create a $\sqrt{\frac{n_{\text{train}}}{(J+1)N_\varepsilon}} \frac{\epsilon}{2}$-cover for $[0, b]$ corresponding to a coordinate in $\mathcal{J}$ and create a $\sqrt{\frac{n_{\text{train}}}{(K-J)(J+1)N_\varepsilon}} \frac{\epsilon}{2}$-cover for $[0, \varepsilon]$ corresponding to a coordinate in $\mathcal{K} \backslash \mathcal{J}$. (Recall that $w_j \leq \varepsilon$ for $j \in \mathcal{K} \backslash \mathcal{J}$.) Then we take the Cartesian product of these $K$ one-dimensional covers and project it to $\mathcal{W}_\varepsilon^*$. This provides a $\sqrt{\frac{n_{\text{train}}}{N_\varepsilon}} \frac{\epsilon}{2}$-cover for $\mathcal{W}_\varepsilon^*$ with a cardinality of

$$\left( \left\lceil \frac{b\sqrt{(J+1)N_\varepsilon}}{\sqrt{n_{\text{train}}}\epsilon} \right\rceil \right)^J \left( \left\lceil \frac{\varepsilon\sqrt{(K-J)(J+1)N_\varepsilon}}{\sqrt{n_{\text{train}}}\epsilon} \right\rceil \right)^{K-J} \leq \left( \frac{2}{\epsilon} \right)^J \left( \left\lceil \frac{2\varepsilon\sqrt{K-J}}{b\epsilon} \right\rceil \right)^{K-J}, \quad (31)$$

where the inequality is because $N_\varepsilon \leq \frac{n_{\text{train}}}{b^2 J}$ by the definition of $N_\varepsilon$. This implies

$$\mathbb{E}[G_{\text{train}}(D^{\text{train}})] \leq 2\widehat{R}_{n_{\text{train}}}(\mathcal{H}, \mathcal{W}_\varepsilon^*) \leq \frac{2C_d}{\sqrt{N_\varepsilon}} \int_0^1 \sqrt{\log \mathcal{N}(\mathcal{H} \times \mathcal{W}_\varepsilon^*; \mathrm{d}; \epsilon)} d\epsilon$$

$$= O\left( \frac{1}{\sqrt{N_\varepsilon}} \int_0^1 \sqrt{(\nu_{\mathcal{H}} + J) \log\left( \frac{1}{\epsilon} \right) + (K-J) \log\left( \left\lceil \frac{2\varepsilon\sqrt{K-J}}{b\epsilon} \right\rceil \right)} d\epsilon \right)$$

$$= O\left( \sqrt{\frac{\nu_{\mathcal{H}} + J}{N_\varepsilon}} \right) + O\left( \sqrt{\frac{K-J}{N_\varepsilon}} \frac{2\varepsilon\sqrt{K-J}}{b} \right) = O\left( \sqrt{\frac{\nu_{\mathcal{H}} + J}{N_\varepsilon}} \right) + O\left( \frac{\varepsilon(K-J)}{b\sqrt{N_\varepsilon}} \right), \quad (32)$$

where the first equality is because

$$\mathcal{N}(\mathcal{H} \times \mathcal{W}_\varepsilon^*; \mathrm{d}; \epsilon) \leq \left( \frac{2C_{\mathcal{H}}}{\epsilon} \right)^{\nu_{\mathcal{H}}} \times \left( \frac{2}{\epsilon} \right)^J \left( \left\lceil \frac{2\varepsilon\sqrt{K-J}}{b\epsilon} \right\rceil \right)^{K-J}$$

according to Assumption 4 and (31) and the second equality is by changing variable $\epsilon$ to $\frac{b\epsilon}{2\varepsilon\sqrt{K-J}}$ in the integral and the fact that $\lceil \frac{1}{\epsilon} \rceil = 0$ when $\epsilon > 1$.

Combining (32) with (26), we have that, with a probability of at least $1 - \delta$,

$$G_{\text{train}}(D^{\text{train}}) \leq O\left( \sqrt{\frac{\nu_{\mathcal{H}} + J + \log(1/\delta)}{N_\varepsilon}} \right) + O\left( \frac{\varepsilon(K-J)}{b\sqrt{N_\varepsilon}} \right).$$

Applying the same argument to $G'_{\text{valid}}(D^{\text{valid}})$, we can show that the same inequality as above holds for $G'_{\text{train}}(D^{\text{train}})$ with a probability of at least $1 - \delta$. By a union bound, we have, with a probability of at least $1 - \delta$

$$\sup_{\theta \in \Theta, w \in \mathcal{W}_\varepsilon^*} \left| \sum_{k=1}^K w_k \left[ \widehat{L}_k(\theta) - L_k(\theta) \right] \right| \leq O\left( \sqrt{\frac{\nu_{\mathcal{H}} + J + \log(1/\delta)}{N_\varepsilon}} \right) + O\left( \frac{\varepsilon(K-J)}{b\sqrt{N_\varepsilon}} \right),$$

which completes the proof (22).

Next we prove (21). Since the proof is similar to (22), we will mainly elaborate the parts that are different. Suppose Assumptions 1, 2 and 4 hold. We define $G_{\text{train}}(D^{\text{train}})$ and $G'_{\text{train}}(D^{\text{train}})$ the same as in (23) and (24) except that $\mathcal{W}_\varepsilon^*$ is replaced by the entire domain $\Delta_K^b$. Following the same proof of (25), we have

$$\left| G_{\text{train}}(D^{\text{train}}) - G_{\text{train}}(D_{j,i}^{\text{train}}) \right| \leq \frac{w_j}{n_j} \leq \frac{b}{n_j} \text{ for all } i \in \{1, 2, \ldots, n_j\} \text{ and } j \in \mathcal{K}.$$

Then the McDiarmid's inequality implies that, with a probability of at least $1 - \delta$,

$$G_{\text{train}}(D^{\text{train}}) \leq \mathbb{E}[G_{\text{train}}(D^{\text{train}})] + \sqrt{\frac{Kb^2 \log(1/\delta)}{2n_{\text{train}}}}. \quad (33)$$

By replacing $\mathcal{W}_\varepsilon^*$ with $\Delta_K^b$ in the proof of (27), we can show that

$$\mathbb{E}[G_{\text{train}}(D^{\text{train}})] \le 2\mathbb{E}\widehat{R}_{n_{\text{train}}}(\mathcal{H}, \Delta_K^b)$$

where $\widehat{R}_{n_{\text{train}}}(\mathcal{H}, \Delta_K^b) := \mathbb{E}\left[\displaystyle\sup_{\theta \in \Theta, w \in \Delta_K^b} \sum_{k=1}^{K} \sum_{i=1}^{n_k} \frac{\sigma_{k,i} w_k}{n_k} l(\theta; z_k^{(i)}) \bigg| D^{\text{train}}\right]$.

Let $N_b$ defined as (20) with $\varepsilon$ replaced by $b$, i.e., $N_b = n_{\text{train}}/(Kb^2)$. Let $M_{\theta,w} := \sqrt{N_b} \sum_{k=1}^{K} \sum_{i=1}^{n_k} \frac{\sigma_{k,i} w_k}{n_k} l(\theta; z_k^{(i)})$ for $\theta \in \Theta$ and $w \in \Delta_K^b$. With $\mathcal{J}$ replaced by $\emptyset$ and $N_b$ replaced by $N_\varepsilon$ in the proof of (28), we have

$$\mathbb{E}\left[\exp\left(\lambda(M_{\theta,w} - M_{\theta',w'})\right)\big|D^{\text{train}}\right] = \exp\left(\frac{\lambda^2}{2}\text{d}^2(\theta, w, \theta', w')\right),$$

where

$$\text{d}(\theta, w, \theta', w') = \sqrt{\sum_{k=1}^{K} \sum_{i=1}^{n_k} \frac{N_b}{n_k^2}\left(w_k l(\theta; z_k^{(i)}) - w_k' l(\theta'; z_k^{(i)})\right)^2} \le \sqrt{\sum_{k \in \mathcal{K}} \frac{N_b b^2}{n_{\text{train}}}} = 1$$

is a pseudo distance metric between $(l(\theta; \cdot), w)$ and $(l(\theta'; \cdot), w')$ in $\mathcal{H} \times \Delta_K^b$. Hence, by Dudley's entropy integral inequality (see Corollary 13.2 in Boucheron et al. (2013)), there exists a universal constant $C_d$ such that

$$\widehat{R}_{n_{\text{train}}}(\mathcal{H}, \Delta_K^b) = \frac{1}{\sqrt{N_b}}\mathbb{E}\left[\sup_{\theta \in \Theta, w \in \Delta_K^b} M_{\theta,w} \bigg| D^{\text{train}}\right] \le \frac{C_d}{\sqrt{N_b}} \int_0^1 \sqrt{\log\left(\mathcal{N}(\mathcal{H} \times \Delta_K^b; \text{d}; \epsilon)\right)} d\epsilon,$$

where $\mathcal{N}(\mathcal{H} \times \Delta_K^b; \text{d}; \epsilon)$ is the $\epsilon$-covering number of $\mathcal{H} \times \Delta_K^b$ w.r.t. $\text{d}$.

Next, we just need to bound $\mathcal{N}(\mathcal{H} \times \mathcal{W}_\varepsilon^*; \text{d}; \epsilon)$. Similar to (29), we can show that

$$\text{d}^2(\theta, w, \theta', w') \le \frac{2N_b}{n_{\text{train}}}\left(\sum_{k \in \mathcal{K}} \sum_{i=1}^{n_{\text{train}}} \frac{b^2}{n_{\text{train}}}\left(l(\theta; z_k^{(i)}) - l(\theta'; z_k^{(i)})\right)^2\right) + \frac{2N_b}{n_{\text{train}}}\|w - w'\|^2.$$

Similar to (30), we define a probability measure $\mathbb{Q} = \frac{N_b}{n_{\text{train}}} \sum_{k \in \mathcal{K}} \sum_{i=1}^{n_{\text{train}}} \frac{b^2}{n_{\text{train}}} \delta_{z_k^{(i)}}$ on $\mathcal{Z}$, where $\delta_{z_k^{(i)}}$ is a point mass at $z_k^{(i)}$. Then, we only need to construct an $\epsilon$-cover for $\mathcal{H} \times \Delta_K^b$ by taking the Cartesian product of an $\frac{\epsilon}{2}$-cover for $\mathcal{H}$ w.r.t. distance metric $\rho_{\mathbb{Q}}$ and a $\sqrt{\frac{n_{\text{train}}}{N_b}}\frac{\epsilon}{2}$-cover for $\Delta_K^b$ w.r.t. the Euclidean distance. According to Assumption 4, the former has a cardinality of $(2C_{\mathcal{H}}/\epsilon)^{\nu_{\mathcal{H}}}$. To construct a $\sqrt{\frac{n_{\text{train}}}{N_b}}\frac{\epsilon}{2}$-cover for $\Delta_K^b$, we first construct a $\sqrt{\frac{n_{\text{train}}}{KN_b}}\frac{\epsilon}{2}$-cover for $[0, b]$, take its $K$-fold Cartesian product, and project it to $\Delta_K^b$. This provides a $\sqrt{\frac{n_{\text{train}}}{N_b}}\frac{\epsilon}{2}$-cover for $\Delta_K^b$ with a cardinality of $\left\lceil \frac{b\sqrt{KN_b}}{\sqrt{n_{\text{train}}}\epsilon} \right\rceil^K = \left\lceil \frac{1}{\epsilon} \right\rceil^K$. This implies $\mathcal{N}(\mathcal{H} \times \Delta_K^b; \text{d}; \epsilon) \le \left(\frac{2C_{\mathcal{H}}}{\epsilon}\right)^{\nu_{\mathcal{H}}} \times \left\lceil \frac{1}{\epsilon} \right\rceil^K$ and thus

$$\mathbb{E}[G_{\text{train}}(D^{\text{train}})] \le 2\widehat{R}_{n_{\text{train}}}(\mathcal{H}, \Delta_K^b) \le \frac{2C_d}{\sqrt{N_b}} \int_0^1 \sqrt{\log \mathcal{N}(\mathcal{H} \times \Delta_K^b; \text{d}; \epsilon)} d\epsilon$$

$$= O\left(\frac{1}{\sqrt{N_b}} \int_0^1 \sqrt{\nu_{\mathcal{H}} \log\left(\frac{1}{\epsilon}\right) + K \log\left(\left\lceil \frac{1}{\epsilon} \right\rceil\right)} d\epsilon\right) = O\left(\sqrt{\frac{\nu_{\mathcal{H}} + K}{N_b}}\right). \tag{34}$$

Combining (34) with (33), we have that, with a probability of at least $1 - \delta$,

$$G_{\text{train}}(D^{\text{train}}) \le O\left(\sqrt{\frac{\nu_{\mathcal{H}} + K + \log(1/\delta)}{N_b}}\right).$$

Applying the same argument to $G'_{\text{valid}}(D^{\text{valid}})$, we can show that the same inequality as above holds for $G'_{\text{train}}(D^{\text{train}})$ with a probability of at least $1 - \delta$. By taking a union bound, we have that, with a probability of at least $1 - \delta$,

$$\sup_{\theta \in \Theta, w \in \Delta_K^b} \left| \sum_{k=1}^K w_k \left[ \widehat{L}_k(\theta) - L_k(\theta) \right] \right| \leq O\left( \sqrt{\frac{\nu_{\mathcal{H}} + K + \log(1/\delta)}{N_b}} \right),$$

which completes the proof of (21) as $N_b = n_{\text{train}}/(Kb^2)$. $\qquad\qquad\square$

## D  PROOFS OF MAIN THEOREMS AND COROLLARIES

In this section, we provide the proofs of Theorem 1, Theorem 2 and Corollary 1.

*Proof of Theorem 1.* By the strong convexity of the loss function and the optimality of $\theta(w)$ and $\widehat{\theta}(w)$ in the inner problems in $(\text{P}_*)$ and $(\widehat{\text{P}})$, we have, for any $w \in \Delta_K^b$,

$$\frac{\mu}{2} \left\| \theta(w) - \widehat{\theta}(w) \right\|^2 \leq \sum_{k=1}^K w_k L_k(\widehat{\theta}(w)) - \sum_{k=1}^K w_k L_k(\theta(w)) \tag{35}$$

$$\frac{\mu}{2} \left\| \theta(w) - \widehat{\theta}(w) \right\|^2 \leq \sum_{k=1}^K w_k \widehat{L}_k(\theta(w)) - \sum_{k=1}^K w_k \widehat{L}_k(\widehat{\theta}(w)). \tag{36}$$

Adding (35) and (36) on both sides leads to, with a probability of at least $1 - \delta$,

$$\mu \left\| \theta(w) - \widehat{\theta}(w) \right\|^2$$

$$\leq \sum_{k=1}^K w_k L_k(\widehat{\theta}(w)) - \sum_{k=1}^K w_k \widehat{L}_k(\widehat{\theta}(w)) + \sum_{k=1}^K w_k \widehat{L}_k(\theta(w)) - \sum_{k=1}^K w_k L_k(\theta(w))$$

$$\leq 2C_a \sqrt{\frac{\nu_{\mathcal{H}} + K + \log(1/\delta)}{n_{\text{train}}/(Kb^2)}}, \quad \forall w \in \Delta_K^b, \tag{37}$$

where the second inequality is because the first conclusion in Lemma 3.

Let $w^* = \text{Proj}_{\mathcal{W}^*}(\widehat{w})$. Then we have, with a probability of $1 - 2\delta$, that

$$F(\widehat{w}) - \min_{w \in \Delta_K^b} F(w) = L_0(\theta(\widehat{w})) - L_0(\theta(w^*))$$

$$\leq L_0(\widehat{\theta}(\widehat{w})) - L_0(\widehat{\theta}(w^*)) + \ell_0 \|\widehat{\theta}(\widehat{w}) - \theta(\widehat{w})\| + \ell_0 \|\widehat{\theta}(w^*) - \theta(w^*)\|$$

$$\leq \widehat{L}_0(\widehat{\theta}(\widehat{w})) - \widehat{L}_0(\widehat{\theta}(w^*)) + 2C_0 \sqrt{\frac{\nu_{\mathcal{H}} + \log(1/\delta)}{n_{\text{valid}}}}$$

$$+ \ell_0 \|\widehat{\theta}(\widehat{w}) - \theta(\widehat{w})\| + \ell_0 \|\widehat{\theta}(w^*) - \theta(w^*)\|$$

$$\leq 2C_0 \sqrt{\frac{\nu_{\mathcal{H}} + \log(1/\delta)}{n_{\text{valid}}}} + 2\ell_0 \sqrt{\frac{2C_a}{\mu}} \left( \frac{\nu_{\mathcal{H}} + K + \log(1/\delta)}{n_{\text{train}}/(Kb^2)} \right)^{\frac{1}{4}}, \tag{38}$$

where the first inequality is because of Assumption 1, the second is due to Lemma 2, and the last is due to (37) and the optimality of $\widehat{w}$ for problem $(\widehat{\text{P}})$. Therefore, we can show that, with a probability of $1 - 2\delta$,

$$L_0(\widehat{\theta}(\widehat{w})) - L_0(\theta^*) = L_0(\widehat{\theta}(\widehat{w})) - L_0(\theta(\widehat{w})) + L_0(\theta(\widehat{w})) - L_0(\theta(w^*))$$

$$\leq \ell_0 \|\widehat{\theta}(\widehat{w}) - \theta(\widehat{w})\| + L_0(\theta(\widehat{w})) - L_0(\theta(w^*))$$

$$\leq 2C_0 \sqrt{\frac{\nu_{\mathcal{H}} + \log(1/\delta)}{n_{\text{valid}}}} + 3\ell_0 \sqrt{\frac{2C_a}{\mu}} \left( \frac{\nu_{\mathcal{H}} + K + \log(1/\delta)}{n_{\text{train}}/(Kb^2)} \right)^{\frac{1}{4}}$$

where the equality is because of Assumption 2, the first inequality is by Assumption 1 and the second by (37) and (38). This completes the proof. $\qquad\square$

*Proof of Theorem 2.* Since Assumption $2'$ implies Assumption 2, the proof and the conclusion of Theorem 1 also hold under the assumptions of Theorem 2. In particular, inequality (38) holds with a probability of $1 - 2\delta$. According to Assumption 3 and (38), we have with a probability of $1 - 2\delta$ that

$$\left[C_r^{-1}\mathrm{Dist}(\widehat{w}, \mathcal{W}^*)\right]^r \le 2C_0\sqrt{\frac{\nu_{\mathcal{H}} + \log(1/\delta)}{n_{\mathrm{valid}}}} + 2\ell_0\sqrt{\frac{2C_a}{\mu}}\left(\frac{\nu_{\mathcal{H}} + K + \log(1/\delta)}{n_{\mathrm{train}}/(Kb^2)}\right)^{\frac{1}{4}}.$$

Applying the fact that $s + t \le (s^{\frac{1}{r}} + t^{\frac{1}{r}})^r$ for any $s > 0$ and $t > 0$ to the right-hand side of the inequality above, we obtain (10) with an appropriately defined $C_w$.

Suppose $\mathrm{Dist}(\widehat{w}, \mathcal{W}^*) \le \varepsilon(n_{\mathrm{valid}}, n_{\mathrm{train}})$, which happens with a probability of $1 - 2\delta$ according to the proof above. We have $\widehat{w} \in \mathcal{W}_\varepsilon^*$ with $\varepsilon = \varepsilon(n_{\mathrm{valid}}, n_{\mathrm{train}})$ according to the definition in (19). We then decompose the optimality gap of the generalization loss as follows

$$
\begin{aligned}
&L_0(\widehat{\theta}(\widehat{w})) - L_0(\theta^*)\\
=\ &\underbrace{L_0(\widehat{\theta}(\widehat{w})) - \sum_{k=1}^{K}\widehat{w}_k L_k(\widehat{\theta}(\widehat{w}))}_{T_1} + \underbrace{\sum_{k=1}^{K}\widehat{w}_k L_k(\widehat{\theta}(\widehat{w})) - \sum_{k=1}^{K}\widehat{w}_k \widehat{L}_k(\widehat{\theta}(\widehat{w}))}_{T_2}\\
&+ \underbrace{\sum_{k=1}^{K}\widehat{w}_k \widehat{L}_k(\widehat{\theta}(\widehat{w})) - \sum_{k=1}^{K}\widehat{w}_k \widehat{L}_k(\theta(\widehat{w}))}_{T_3} + \underbrace{\sum_{k=1}^{K}\widehat{w}_k \widehat{L}_k(\theta(\widehat{w})) - \sum_{k=1}^{K}\widehat{w}_k L_k(\theta(\widehat{w}))}_{T_4}\\
&+ \underbrace{\sum_{k=1}^{K}\widehat{w}_k L_k(\theta(\widehat{w})) - \sum_{k=1}^{K}\widehat{w}_k L_k(\theta(w^*))}_{T_5} + \underbrace{\sum_{k=1}^{K}\widehat{w}_k L_k(\theta^*) - L_0(\theta^*)}_{T_6}.
\end{aligned}
\tag{39}
$$

It is clear that $T_3 \le 0$ and $T_5 \le 0$ by the optimality of $\widehat{\theta}(\widehat{w})$ and $\theta(\widehat{w})$ in $(\widehat{\mathrm{P}})$ and $(\mathrm{P}_*)$, respectively. Moreover, by Assumption $2'$, we have $w_j^* = 0$ for $w \in \mathcal{W}^*$ and $j \in \mathcal{K}\backslash\mathcal{J}$. Using the fact that $\widehat{w} \in \mathcal{W}_\varepsilon^*$, we have $\sum_{k\in\mathcal{K}\backslash\mathcal{J}}\widehat{w}_k^2 \le \mathrm{Dist}^2(\widehat{w}, \mathcal{W}^*) \le \varepsilon^2$, which implies

$$
\begin{aligned}
T_1 &= \sum_{k=1}^{K}\widehat{w}_k\left(L_0(\widehat{\theta}(\widehat{w})) - L_k(\widehat{\theta}(\widehat{w}))\right) = \sum_{k\in\mathcal{K}\backslash\mathcal{J}}\widehat{w}_k\left(L_0(\widehat{\theta}(\widehat{w})) - L_k(\widehat{\theta}(\widehat{w}))\right)\\
&\le \sqrt{\sum_{k\in\mathcal{K}\backslash\mathcal{J}}\widehat{w}_k^2}\sqrt{\max_{\theta\in\Theta}\sum_{k\in\mathcal{K}\backslash\mathcal{J}}\left(L_0(\theta) - L_k(\theta)\right)^2} \le \varepsilon\sqrt{\max_{\theta\in\Theta}\sum_{k\in\mathcal{K}\backslash\mathcal{J}}\left(L_0(\theta) - L_k(\theta)\right)^2}.
\end{aligned}
$$

Using a similar argument, we can also show

$$T_6 \le \varepsilon\sqrt{\max_{\theta\in\Theta}\sum_{k\in\mathcal{K}\backslash\mathcal{J}}\left(L_0(\theta) - L_k(\theta)\right)^2}.$$

According to Lemma 3 and the fact that $\widehat{w} \in \mathcal{W}_\varepsilon^*$, we have that, with a probability of at least $1 - \delta$,

$$T_2,\ T_4 \le C_a'\left(\sqrt{\frac{\nu_{\mathcal{H}} + J + \log(1/\delta)}{N_\varepsilon}} + \frac{\varepsilon(K-J)}{b\sqrt{N_\varepsilon}}\right).$$

Note that $G = \sqrt{\max_{\theta\in\Theta}\sum_{k\in\mathcal{K}\backslash\mathcal{J}}\left(L_0(\theta) - L_k(\theta)\right)^2}$ under Assumption $2'$. Applying the upper bounds of the six terms to (39) and taking a union bound, we can show that

$$L_0(\widehat{\theta}(\widehat{w})) - L_0(\theta^*) \le 2C_a'\left(\sqrt{\frac{\nu_{\mathcal{H}} + J + \log(1/\delta)}{N_\varepsilon}} + \frac{\varepsilon(K-J)}{b\sqrt{N_\varepsilon}}\right) + 2\varepsilon(n_{\mathrm{valid}}, n_{\mathrm{train}})G$$

with a probability of at least $1 - 3\delta$, which completes the proof. $\qquad\square$

Before we prove Corollary 1, we first present another corollary of Theorem 2 where we can see the impact of $b$ more clearly.

**Corollary 2.** *Suppose the assumptions of Theorem 2 hold and $n_{valid}$ and $n_{train}$ are large enough such that $\varepsilon(n_{valid}, n_{train})$ defined in (10) satisfies $\varepsilon(n_{valid}, n_{train}) \leq \frac{b\sqrt{J}}{K-J}$. With a probability of at least $1 - 3\delta$, we have*

$$
L_0(\widehat{\theta}(\widehat{w})) - L_0(\theta^*) \leq O\left(\sqrt{\frac{\nu_{\mathcal{H}} + J + \log(1/\delta)}{n_{train}/(Jb^2)}}\right)
$$
$$
+ G \cdot O\left(\frac{\nu_{\mathcal{H}} + \log(1/\delta)}{n_{valid}}\right)^{\frac{1}{2r}} + G \cdot O\left(\frac{\nu_{\mathcal{H}} + K + \log(1/\delta)}{n_{train}/(Kb^2)}\right)^{\frac{1}{4r}}, \tag{40}
$$

*Proof.* Theorem 2 guarantees that (11) holds with a high probability. When $\varepsilon(n_{\text{valid}}, n_{\text{train}}) \leq \frac{b\sqrt{J}}{K-J}$, the second term on the right-hand side of (11) can be merged with the first term. Also, we have $N_\varepsilon = \frac{n_{\text{train}}}{b^2 J + \varepsilon^2(n_{\text{valid}}, n_{\text{train}})(K-J)} \geq \frac{n_{\text{train}}}{b^2 J + b^2 J/(K-J)} \geq \frac{n_{\text{train}}}{2b^2 J}$, where the last inequality is because $K - J \geq 1$. As a result, the first two terms in (11) together has the order of

$$
O\left(\sqrt{\frac{\nu_{\mathcal{H}} + J + \log(1/\delta)}{n_{\text{train}}/(Jb^2)}}\right).
$$

Then (40) is proved by applying the definition of $\varepsilon(n_{\text{valid}}, n_{\text{train}})$ to the third term on the right-hand side of (11). □

Suppose $n_{\text{train}} \gg n_{\text{valid}}$. The first term on the right-hand side of (40) is smaller than the entire right-hand side of (7). If, in addition, $G = o(1/n_{\text{valid}}^{\frac{1}{2} - \frac{1}{2r}})$ and $G = o(1/n_{\text{train}}^{\frac{1}{4} - \frac{1}{4r}})$, the other two terms in (40) are also smaller than the two terms in (7), respectively, so the bound in (40) is tighter than (7).

*Proof of Corollary 1.* Corollary 1 is directly from Corollary (2) by only keeping $G$, $n_{\text{train}}$ and $n_{\text{test}}$ in the order of magnitude given in (40). □

# E  GENERALIZATION PERFORMANCE BY TRAINING LOCALLY AND TRAINING WITH EQUALLY WEIGHTED NODES

In this section, we first consider a model locally trained only with data $D^{\text{valid}}$ in node 0, namely,

$$
\widehat{\theta}_{\text{valid}} \in \arg\min_{\theta \in \Theta} \widehat{L}_0(\theta) \tag{41}
$$

where $\widehat{L}_0$ is given in (3). The generalization bound of $\widehat{\theta}_{\text{valid}}$ is well-known, so we omit the proof but directly give the result.

**Proposition 1.** *Suppose Assumptions 1 and 4 hold. There exists a universal constant $C_0 > 0$ such that, with a probability of at least $1 - \delta$,*

$$
L_0(\widehat{\theta}_{\text{valid}}) - L_0(\theta^*) \leq 2C_0 \sqrt{\frac{\nu_{\mathcal{H}} + \log(1/\delta)}{n_{valid}}}.
$$

For the purpose of theoretical comparison, we also consider a model trained only with data $D^{\text{train}}$ distributed over equally weighted nodes, namely,

$$
\widehat{\theta}_{\text{equal}} \in \arg\min_{\theta \in \Theta} \frac{1}{K} \sum_{k=1}^{K} \widehat{L}_k(\theta), \tag{42}
$$

where $\widehat{L}_k$ is defined as in (2). It is easy to construct an example where each $p_k$ with $k \in \mathcal{K}\backslash\mathcal{J}$ is significantly different from $p_0$ so that $L_0(\widehat{\theta}_{\text{equal}})$ does not convergence to $L_0(\theta^*)$ as $n_{\text{train}}$ goes to infinity. Motivated by Corollary 1 and the discussion afterwards, it will be interesting to show the generalization bound of $\widehat{\theta}_{\text{equal}}$ when each $p_k$ with $k \in \mathcal{K}\backslash\mathcal{J}$ is similar to $p_0$ with a small $G$ defined in (9).

**Proposition 2.** *Suppose Assumptions [1], [2'] and [4] hold. There exists a universal constant $C_a > 0$ such that, with a probability of at least $1 - 3\delta$,*

$$L_0(\widehat{\theta}_{\text{equal}}) - L_0(\theta^*) \leq 2C_a \sqrt{\frac{\nu_{\mathcal{H}} + K + \log(1/\delta)}{Kn_{train}}} + \frac{2\sqrt{K-J}}{K}G,$$

*where $G$ is defined in ([9]).*

*Proof.* We first define

$$\theta_{\text{equal}} \in \underset{\theta \in \Theta}{\arg\min} \frac{1}{K} \sum_{k=1}^{K} L_k(\theta). \tag{43}$$

We first decompose the optimality gap of the generalization loss as follows

$$
\begin{aligned}
&L_0(\widehat{\theta}_{\text{equal}}) - L_0(\theta^*)\\
={}& \underbrace{L_0(\widehat{\theta}_{\text{equal}}) - \frac{1}{K}\sum_{k=1}^{K} L_k(\widehat{\theta}_{\text{equal}})}_{T_1} + \underbrace{\frac{1}{K}\sum_{k=1}^{K} L_k(\widehat{\theta}_{\text{equal}}) - \frac{1}{K}\sum_{k=1}^{K} \widehat{L}_k(\widehat{\theta}_{\text{equal}})}_{T_2}\\
&+ \underbrace{\frac{1}{K}\sum_{k=1}^{K} \widehat{L}_k(\widehat{\theta}_{\text{equal}}) - \frac{1}{K}\sum_{k=1}^{K} \widehat{L}_k(\theta_{\text{equal}})}_{T_3} + \underbrace{\frac{1}{K}\sum_{k=1}^{K} \widehat{L}_k(\theta_{\text{equal}}) - \frac{1}{K}\sum_{k=1}^{K} L_k(\theta_{\text{equal}})}_{T_4}\\
&+ \underbrace{\frac{1}{K}\sum_{k=1}^{K} L_k(\theta_{\text{equal}}) - \frac{1}{K}\sum_{k=1}^{K} L_k(\theta^*)}_{T_5} + \underbrace{\frac{1}{K}\sum_{k=1}^{K} L_k(\theta^*) - L_0(\theta^*)}_{T_6}
\end{aligned} \tag{44}
$$

It is clear that $T_3 \leq 0$ and $T_5 \leq 0$ by the optimality of $\widehat{\theta}_{\text{equal}}$ and $\theta_{\text{equal}}$ in ([42]) and ([43]), respectively. Moreover, we have

$$
\begin{aligned}
T_1 ={}& \frac{1}{K}\sum_{k=1}^{K}\left(L_0(\widehat{\theta}_{\text{equal}}) - L_k(\widehat{\theta}_{\text{equal}})\right) = \sum_{k \in \mathcal{K}\setminus\mathcal{J}}\frac{1}{K}\left(L_0(\widehat{\theta}_{\text{equal}}) - L_k(\widehat{\theta}_{\text{equal}})\right)\\
\leq{}& \sqrt{\frac{K-J}{K^2}}\sqrt{\max_{\theta \in \Theta}\sum_{k \in \mathcal{K}\setminus\mathcal{J}}\left(L_0(\theta) - L_k(\theta)\right)^2}.
\end{aligned}
$$

Using a similar argument, we can also show

$$T_6 \leq \sqrt{\frac{K-J}{K^2}}\sqrt{\max_{\theta \in \Theta}\sum_{k \in \mathcal{K}\setminus\mathcal{J}}\left(L_0(\theta) - L_k(\theta)\right)^2}.$$

Since Assumption [2'] implies Assumption [2], by the first statement of Lemma [3] with $b = \frac{1}{K}$, we have, with a probability of at least $1 - \delta$, that

$$T_2,\ T_4 \leq C_a\sqrt{\frac{\nu_{\mathcal{H}} + K + \log(1/\delta)}{Kn_{\text{train}}}}.$$

Applying the upper bounds of the six terms to ([44]) and a taking union bound, we have

$$L_0(\widehat{\theta}_{\text{equal}}) - L_0(\theta^*) \leq 2C_a\sqrt{\frac{\nu_{\mathcal{H}} + K + \log(1/\delta)}{Kn_{\text{train}}}} + \frac{2\sqrt{K-J}}{K}G$$

with a probability of at least $1 - 3\delta$, which completes the proof. $\qquad\square$

Note that the bound in Proposition [2] is strictly worse than the one we showed in Corollary [1] for any value of $G$. In fact, the former is $O(1/\sqrt{n_{\text{train}}} + G)$ and the latter is $O(1/\sqrt{n_{\text{train}}}) + o(G)$

## F COMMUNICATION COMPLEXITY OF ALGORITHM 2 AND EXTENSION TO NON-CONVE CASE

In this section, we present the communication complexity of Algorithm 2 for convex problems as well as the corresponding algorithm and complexity for non-convex problems. To do so, we first present the convergence property of Algorithm 1, which is originally established by Gorbunov et al. (2021). Then, we combine the analysis by Gorbunov et al. (2021) and Ghadimi & Wang (2018) with some minor but necessary modifications, for example, to allow for a generic weight $w$ instead of the uniform weight in Gorbunov et al. (2021), and to handle the approximation error between $\nabla \widehat{F}(w)$ and $\bar{\nabla} \widehat{F}(w)$, which is a little different from the one considered in Ghadimi & Wang (2018).

### F.1 CONVERGENCE PROPERTY OF ALGORITHM 1

As mentioned in Section 4, we need to solve subproblems (1) and (15) in Algorithm 2, both of which are instances of (16). Because of Assumption 1, problem (16) in these two cases satisfies the following assumption with $\ell_1$ and $\mu$ exactly the same as the $\ell_1$ and $\mu$ in Assumption 1.

**Assumption 5.** $f_{k,i}(x)$ is convex, $\nabla f_{k,i}(x)$ is $\ell_1$-Lipschitz continuous and $f_k(x)$ is $\mu$-strongly convex for $i = 1, \ldots, n_k$ and $k = 1, 2, \ldots, K$.

A unified analysis is provided in Gorbunov et al. (2021) for a large class of FL methods including Local-SVRG given in Algorithm 1. The following proposition is obtained by applying Theorem 2.1 in Gorbunov et al. (2021) to Local-SVRG under our setting after minor modifications. It characterizes the convergence property of Algorithm 1. We omit its proof because it is the almost the same as the proof of Theorem G.7 in Gorbunov et al. (2021).

**Proposition 3.** *Suppose Assumption 5 holds for (16) and $\gamma \leq \gamma_0$ with $\gamma_0$ defined in (17). Algorithm 1 guarantees*

$$\mathbb{E}\left[f(\overline{x}^{(T)}) - f(x^*)\right] \leq \frac{1}{\gamma}\left(1 - \gamma\mu\right)^{T+1}\left(4 + \frac{32\gamma^2\ell_1^2}{3q} + 30e\gamma^3\ell_1^3(\tau-1)\frac{2+q}{q}\right)\left\|x^{(0)} - x^*\right\|^2$$

$$+ \frac{45e}{2}\ell_1\gamma^2(\tau-1)^2\sum_{k=1}^{K} w_k\left\|\nabla f_k(x^*)\right\|^2, \tag{45}$$

*where $x^*$ be the optimal solution of (16).*

### F.2 COMMUNICATION COMPLEXITY OF ALGORITHM 2

To analyze the complexity of Algorithm 2, we needs to bound the error of the approximate gradient of $\widehat{F}$, namely, the quantity

$$\left\|\bar{\nabla} \widehat{F}(w_{\text{md}}^{(s)}) - \nabla \widehat{F}(w_{\text{md}}^{(s)})\right\|, \quad s = 0, 1, \ldots,$$

and then the convergence analysis in Ghadimi & Wang (2018) can be directly applied. This error depends the suboptimality of $\theta^{(s)}$ and $h^{(s)}$ in iteration $s$ of Algorithm 2, which can be characterized using Proposition 3. To do so, we first bound $\|\nabla f_k(x^*)\|$ and $\|x^{(0)} - x^*\|$ appearing in Proposition 3 for these two instances. For simplicity, we assume Local-SVRG is initialized at $x^{(0)} = 0$ when it is applied to any instance of (16).

Suppose $f_{k,i}(\theta) = l(\theta; z_k^{(i)}), i = 1, \ldots, n_k, k = 1, \ldots, K$ and $x^*$ is the optimal solution of (16), namely, $x^* = \widehat{\theta}(w)$. Because of Assumption 1, we have that $\left\|\nabla f_k(x^*)\right\| \leq \ell_0$ for any $k$ in any iteration $s$ of Algorithm 2 and $x^* = \widehat{\theta}(w)$ is a continuous of $w$ on $\Delta_K^b$, which means $\left\|x^{(0)} - x^*\right\|^2 = \left\|x^*\right\|^2 \leq \max_{w \in \Delta_K^b} \|\widehat{\theta}(w)\|^2$ in any iteration $s$ of Algorithm 2.

Suppose $f_{k,i}(h) = \frac{1}{2}h^\top \nabla^2 l(\theta^{(s)}; z_k^{(i)})h - h^\top \nabla \widehat{L}_0(\theta^{(s)}), i = 1, \ldots, n_k, k = 1, \ldots, K$. The optimal solution of (16) in this case is

$$x^* = \left(\sum_{k=1}^{K} w_k^{(s)} \nabla^2 \widehat{L}_k(\theta^{(s)})\right)^{-1} \nabla \widehat{L}_0(\theta^{(s)})$$

which means $\|x^{(0)} - x^*\|^2 = \|x^*\|^2 \leq \frac{1}{\mu^2}\|\nabla \widehat{L}_0(\theta^{(s)})\|^2 \leq \frac{\ell_0^2}{\mu^2}$ because of Assumption 1. Moreover,

$$\nabla f_k(x^*) = \nabla^2 \widehat{L}_k(\theta^{(s)}) \left(\sum_{k=1}^{K} w_k^{(s)} \nabla^2 \widehat{L}_k(\theta^{(s)})\right)^{-1} \nabla \widehat{L}_0(\theta^{(s)}) - \nabla \widehat{L}_0(\theta^{(s)}),$$

so $\|\nabla f_k(x^*)\| \leq \frac{\ell_1 \ell_0}{\mu} + \ell_0$ for any $k$ and $s$ by Assumption 1.

Since $f$ is $\mu$-strongly convex, we have $\frac{\mu}{2}\|\overline{x}^{(T)} - x^*\|^2 \leq f(\overline{x}^{(T)}) - f(x^*)$. With this inequality and the discussion observations, we can derive from (45) that

$$\mathbb{E}\big[\|\overline{x}^{(T)} - x^*\|^2\big] \leq \frac{2}{\gamma\mu}\left(1 - \gamma\mu\right)^{T+1}\left(4 + \frac{32\gamma^2\ell_1^2}{3q} + 30e\gamma^3\ell_1^3(\tau - 1)\frac{2+q}{q}\right) \cdot \max\left\{R^2, \frac{\ell_0^2}{\mu^2}\right\}$$

$$+ \frac{45e}{\mu}\ell_1\gamma^2(\tau - 1)^2(\frac{\ell_1}{\mu} + 1)^2\ell_0^2 \tag{46}$$

with $R := \max_{w \in \Delta_K^b} \|\widehat{\theta}(w)\|$ when Local-SVRG is applied to either (1) or (15) in any iteration of Algorithm 2.

The following lemma bounds the error of the approximate gradient for $\widehat{F}$.

**Lemma 4.** *Suppose Assumption 5 holds and $\gamma \leq \gamma_0$ with $\gamma_0$ defined in (17). We have*

$$\mathbb{E}\big[\|\nabla\widehat{F}(w_{md}^{(s)}) - \bar{\nabla}\widehat{F}(w_{md}^{(s)})\|^2\big] \leq \frac{C_1}{\gamma}\left(1 - \gamma\mu\right)^{T_s+1}\left(C_2 + C_3(\tau - 1)\right) + C_4\gamma^2(\tau - 1)^2, \tag{47}$$

*where $C_1$, $C_2$ and $C_3$ are constants that depend on $\ell_0$, $\ell_1$, $\ell_2$, $\mu$, $R$, $q$ and $K$ but not $\tau$, $T_s$ and $\gamma$. Consequently,*

- *when $\tau = 1$, $\gamma = \gamma_0$ and $T_s = \frac{1}{\gamma_0\mu}\ln\left(\frac{C_1 C_2(s+1)^4}{\gamma_0}\right)$ OR*

- *when $\tau > 1$, $\gamma = \frac{1}{M_s}$ and $T_s = \mu^{-1}M_s\ln\left(M_s^3\right)$,*

*where*

$$M_s = \max\left\{\frac{1}{\gamma_0}, (s+1)^2\sqrt{[C_1(C_2 + C_3(\tau - 1)) + C_4(\tau - 1)^2]}\right\},$$

*we have $\mathbb{E}\big[\|\nabla\widehat{F}(w_{md}^{(s)}) - \bar{\nabla}\widehat{F}(w_{md}^{(s)})\|^2\big] \leq \frac{1}{(s+1)^4}$.*

*Moreover,*

- *when $\tau = 1$, $\gamma = \gamma_0$ and $T_s = \frac{1}{\gamma_0\mu}\ln\left(\frac{C_1 C_2(s+1)^2}{\gamma_0}\right)$ OR*

- *when $\tau > 1$, $\gamma = \frac{1}{M_s'}$ and $T_s = \mu^{-1}M_s'\ln\left(M_s'^3\right)$,*

*where*

$$M_s' = \max\left\{\frac{1}{\gamma_0}, (s+1)\sqrt{[C_1(C_2 + C_3(\tau - 1)) + C_4(\tau - 1)^2]}\right\},$$

*we have $\mathbb{E}\big[\|\nabla\widehat{F}(w_{md}^{(s)}) - \bar{\nabla}\widehat{F}(w_{md}^{(s)})\|^2\big] \leq \frac{1}{(s+1)^2}$.*

*Proof.* Let $\widetilde{\nabla}\widehat{F}(w_{md}^{(s)}) = (\widetilde{\nabla}_k\widehat{F}(w_{md}^{(s)}))_{k=1,\ldots,K}$, where

$$\widetilde{\nabla}_k\widehat{F}(w_{md}^{(s)}) = -\nabla\widehat{L}_k(\theta^{(s)})^\top\left(\sum_{k=1}^{K} w_{md,k}^{(s)}\nabla^2\widehat{L}_k(\theta^{(s)})\right)^{-1}\nabla\widehat{L}_0(\theta^{(s)}). \tag{48}$$

Recall that

$$\bar{\nabla}_k\widehat{F}(w_{md}^{(s)}) = -\nabla\widehat{L}_k(\theta^{(s)})^\top h^{(s)}.$$

We then obtained from (46) that

$$
\mathbb{E}\big[\big\|\widetilde{\nabla}\widehat{F}(w_{\mathrm{md}}^{(s)}) - \bar{\nabla}\widehat{F}(w_{\mathrm{md}}^{(s)})\big\|^2\big]
$$
$$
\leq K\ell_0^2 \mathbb{E}\Bigg[\bigg\| h^{(s)} - \bigg(\sum_{k=1}^{K} w_{\mathrm{md},k}^{(s)} \nabla^2 \widehat{L}_k(\theta^{(s)})\bigg)^{-1} \nabla\widehat{L}_0(\theta^{(s)})\bigg\|^2\Bigg]
$$
$$
\leq \frac{2K\ell_0^2}{\gamma\mu}\big(1-\gamma\mu\big)^{T_s+1}\bigg(4 + \frac{32\gamma_0^2\ell_1^2}{3q} + 30e\gamma_0^3\ell_1^3(\tau-1)\frac{2+q}{q}\bigg)\cdot\max\Big\{R^2, \frac{\ell_0^2}{\mu^2}\Big\}
$$
$$
+ \frac{45eK\ell_0^2}{\mu}\ell_1\gamma^2(\tau-1)^2(\frac{\ell_1}{\mu}+1)^2\ell_0^2 \tag{49}
$$

According to Lemma 2.2 in Ghadimi & Wang (2018) and (46), we have

$$
\mathbb{E}\big[\big\|\widetilde{\nabla}\widehat{F}(w_{\mathrm{md}}^{(s)}) - \nabla\widehat{F}(w_{\mathrm{md}}^{(s)})\big\|^2\big]
$$
$$
\leq K\bigg(\frac{2\ell_0\ell_1}{\mu} + \frac{\ell_2\ell_0^2}{\mu^2}\bigg)^2 \mathbb{E}\big[\|\theta^{(s)} - \widehat{\theta}(w_{\mathrm{md}}^{(s)})\|^2\big]
$$
$$
\leq K\bigg(\frac{2\ell_0\ell_1}{\mu} + \frac{\ell_2\ell_0^2}{\mu^2}\bigg)^2 \frac{2}{\gamma\mu}\big(1-\gamma\mu\big)^{T_s+1}\bigg(4 + \frac{32\gamma_0^2\ell_1^2}{3q} + 30e\gamma_0^3\ell_1^3(\tau-1)\frac{2+q}{q}\bigg)\cdot\max\Big\{R^2, \frac{\ell_0^2}{\mu^2}\Big\}
$$
$$
+ K\bigg(\frac{2\ell_0\ell_1}{\mu} + \frac{\ell_2\ell_0^2}{\mu^2}\bigg)^2 \frac{45e}{\mu}\ell_1\gamma^2(\tau-1)^2(\frac{\ell_1}{\mu}+1)^2\ell_0^2 \tag{50}
$$

Combining (49) and (50) by the triangle inequity leads to (47).

When $\tau = 1$, $\gamma = \gamma_0$ and $T_s = \frac{1}{\gamma_0\mu}\ln\big(\frac{C_1 C_2(s+1)^4}{\gamma_0\mu}\big)$, it is easy to show that $\mathbb{E}\big[\big\|\nabla\widehat{F}(w_{\mathrm{md}}^{(s)}) - \bar{\nabla}\widehat{F}(w_{\mathrm{md}}^{(s)})\big\|^2\big] \leq \frac{1}{(s+1)^4}$.

Suppose $\tau > 1$, $\gamma = \frac{1}{M_s}$ and $T_s = \mu^{-1}M_s\ln\big(M_s^3\big)$. We have

$$
\begin{aligned}
\mathbb{E}\big[\big\|\nabla\widehat{F}(w_{\mathrm{md}}^{(s)}) - \bar{\nabla}\widehat{F}(w_{\mathrm{md}}^{(s)})\big\|^2\big] &\leq \frac{C_1}{\gamma}\big(1-\gamma\mu\big)^{T_s+1}\big(C_2 + C_3(\tau-1)\big) + C_4\gamma^2(\tau-1)^2 \\
&\leq \exp\Big(-\frac{\mu T_s}{M_s}\Big) M_s C_1\big(C_2 + C_3(\tau-1)\big) + \frac{C_4(\tau-1)^2}{M_s^2} \\
&\leq \exp\big(-\ln(M_s^3)\big) M_s C_1\big(C_2 + C_3(\tau-1)\big) + \frac{C_4(\tau-1)^2}{M_s^2} \\
&\leq \frac{C_1\big(C_2 + C_3(\tau-1)\big)}{M_s^2} + \frac{C_4(\tau-1)^2}{M_s^2} \leq \frac{1}{(s+1)^4}.
\end{aligned}
$$

The conclusion with $\mathbb{E}\big[\big\|\nabla\widehat{F}(w_{\mathrm{md}}^{(s)}) - \bar{\nabla}\widehat{F}(w_{\mathrm{md}}^{(s)})\big\|^2\big] \leq \frac{1}{(s+1)^2}$ can be proved in the same way except that $(s+1)^4$ must be changed to $(s+1)^2$, and thus we omit the proof. $\qquad\square$

The complexity of Algorithm 2 when $\widehat{F}(w)$ is convex can be showed using the proof in Ghadimi & Wang (2018) with their gradient approximation error replaced by the one in Lemma 4.

*Proof of Theorem 3.* According to Lemma 2.2 in Ghadimi & Wang (2018), $\widehat{F}(w)$ is $\ell_F$-smooth with $\ell_F$ defined in (12). Let $E_s := \big\|\nabla\widehat{F}(w_{\mathrm{md}}^{(s)}) - \bar{\nabla}\widehat{F}(w_{\mathrm{md}}^{(s)})\big\|$ According to (2.51) in Ghadimi & Wang

(2018), we have

$$
\begin{aligned}
\widehat{F}(w_{\mathrm{ag}}^{(s+1)}) \leq & \frac{s}{s+2}\widehat{F}(w_{\mathrm{ag}}^{(s)}) + \frac{2}{s+2}\widehat{F}(\widehat{w}) + \frac{16}{\eta(s+1)(s+2)}\left(\|\widehat{w}-w^{(s)}\|^2 - \|\widehat{w}-w^{(s+1)}\|^2\right) \\
& + \frac{2}{s+2}\|\widehat{w}-w^{(s+1)}\|E_s + \frac{\eta}{2}E_s^2 \\
\leq & \frac{s}{s+2}\widehat{F}(w_{\mathrm{ag}}^{(s)}) + \frac{2}{s+2}\widehat{F}(\widehat{w}) + \frac{16}{\eta(s+1)(s+2)}\left(\|\widehat{w}-w^{(s)}\|^2 - \|\widehat{w}-w^{(s+1)}\|^2\right) \\
& + \frac{4}{s+2}E_s + \frac{\eta}{2}E_s^2,
\end{aligned}
$$

where the second inequality is because $\|\widehat{w}-w^{(s+1)}\| \leq 2$ as both $\widehat{w}$ and $w^{(s+1)}$ are on a simplex. Subtracting $\widehat{F}(\widehat{w})$ from both sides of the inequality above and dividing both sides by $\frac{2}{(s+1)(s+2)}$, we have

$$
\begin{aligned}
& \frac{(s+1)(s+2)}{2}\left[\widehat{F}(w_{\mathrm{ag}}^{(s+1)}) - \widehat{F}(\widehat{w})\right] \\
\leq & \frac{s(s+1)}{2}\left[\widehat{F}(w_{\mathrm{ag}}^{(s)}) - \widehat{F}(\widehat{w})\right] + \frac{8}{\eta}\left(\|\widehat{w}-w^{(s)}\|^2 - \|\widehat{w}-w^{(s+1)}\|^2\right) \\
& + 2(s+1)E_s + \frac{\eta(s+1)(s+2)}{4}E_s^2.
\end{aligned}
$$

Summing up this inequality for $s = 0, 1, \ldots, S-1$ gives

$$
\frac{S(S+1)}{2}\mathbb{E}\left[\widehat{F}(w_{\mathrm{ag}}^{(S)}) - \widehat{F}(\widehat{w})\right] \leq \frac{16}{\eta} + \sum_{s=0}^{S-1}2(s+1)\sqrt{\mathbb{E}\left[E_s^2\right]} + \sum_{s=0}^{S-1}\frac{\eta(s+1)(s+2)}{4}\mathbb{E}\left[E_s^2\right],
$$

which, when $\mathbb{E}\left[E_s^2\right] \leq \frac{1}{(s+2)^4}$, implies

$$
\mathbb{E}\left[\widehat{F}(w_{\mathrm{ag}}^{(S)}) - \widehat{F}(\widehat{w})\right] \leq \frac{32}{\eta S(S+1)} + \frac{4\log(S)}{S(S+1)} + \frac{\pi^2\eta}{12S(S+1)}.
$$

This means, as long as $\mathbb{E}\left[E_s^2\right] \leq \frac{1}{(s+2)^4}$, Algorithm 2 finds an $\epsilon$-optimal solution of $(\widehat{\mathrm{P}})$ in $\tilde{O}(\epsilon^{-0.5})$ iterations.

Let $A_1 = C_1C_2$, $A_2 = C_1C_3$ and $A_3 = C_1C_4$ with $C_1$, $C_2$, $C_3$ and $C_4$ defined in Lemma 4.

Suppose $\tau = 1$, $\gamma = \gamma_0$ and $T_s = \frac{1}{\gamma_0\mu}\ln\left(\frac{A_1(s+1)^4}{\gamma_0}\right)$. By Lemma 4, we have $\mathbb{E}\left[E_s^2\right] \leq \frac{1}{(s+2)^4}$. In iteration $s$ of Algorithm 2, the total number of rounds of communication needed in Local-SVRG is $O(T_s) = O(\ln(s))$ so that means the total number of rounds is $\tilde{O}(\epsilon^{-0.5})$.

Suppose $\tau > 1$, $\gamma = \frac{1}{M_s}$ and $T_s = \mu^{-1}M_s\ln\left(M_s^3\right)$. By Lemma 4, we have $\mathbb{E}\left[E_s^2\right] \leq \frac{1}{(s+2)^4}$. In iteration $s$ of Algorithm 2, the total number of rounds of communication needed in Local-SVRG is $O(T_s/\tau) = O(s^2)$ so that means the total number of rounds is $\tilde{O}(\epsilon^{-1.5})$. $\qquad\square$

## F.3 ALGORITHM AND COMMUNICATION COMPLEXITY FOR NON-CONVEX $\widehat{F}$

When $\widehat{F}$ in $(\widehat{\mathrm{P}})$ is non-convex, we no long expect any algorithm to find an $\epsilon$-optimal solution and change our goal to finding an $\epsilon$-*stationary* point of $(\widehat{\mathrm{P}})$, which is defined as a solution $\bar{w} \in \Delta_K^b$ satisfying

$$
\eta^{-1}\left\|\bar{w} - \mathrm{Proj}_{\Delta_K^b}(\bar{w} - \eta\nabla\widehat{F}(\bar{w}))\right\| \leq \epsilon.
$$

for some $\eta > 0$. There exist multiple numerical techniques for finding an $\epsilon$-stationary, among which the proximal gradient method is the simplest one. When the gradient can only be computed inexactly, there exist studies on the iteration complexity of the proximal gradient method for finding an $\epsilon$-stationary point, including Ghadimi & Wang (2018) for bilevel optimization and Gu et al. (2018) for a general problem. We will simply apply the proximal gradient method to $(\widehat{\mathrm{P}})$ using the approximate gradient $\bar{\nabla}\widehat{F}(w)$ in (14). We formally present this approach in Algorithm 3. Again, the center is node 0, i.e., the node where $D^{\mathrm{valid}}$ is stored.

---

**Algorithm 3:** Federated Learning Method for Bilevel Optimization ($\widehat{P}$) (Non-Convex Case)

---

1    **Input**: initial weight $w^{(0)}$, learning rate $\eta$, training data $D_k^{\text{train}}$ for $k \in \mathcal{K}$, validation data $D^{\text{valid}}$, the number of outer iterations $S$, parameters $(\gamma, \tau, q)$ for `Local-SVRG`, and the number of inner iterations $T_s$ for $s = 0, \ldots, S-1$

2    **for** $s = 0, 1, \ldots, S-1$ **do**

3       Compute $\bar{\nabla}\widehat{F}(w^{(s)})$ as follows:

4          Set $f_{k,i}(\theta) = l(\theta; z_k^{(i)}), \quad i = 1, \ldots, n_k, \quad k = 1, \ldots, K$

5          Compute $\theta^{(s)} = $ `Local-SVRG`$(\{f_{k,i}\}, w^{(s)}, \theta^{(s-1)}, \gamma, \tau, q, T_s)$ and send it to each node.

6          Compute $\nabla\widehat{L}_0(\theta^{(s)})$ at center and send it to each node.

7          Set $f_{k,i}(h) = \frac{1}{2}h^\top \nabla^2 l(\theta^{(s)}; z_k^{(i)})h - h^\top \nabla\widehat{L}_0(\theta^{(s)}), \quad i = 1, \ldots, n_k, \quad k = 1, \ldots, K$

8          Compute $h^{(s)} = $ `Local-SVRG`$(\{f_{k,i}\}, w^{(s)}, \nabla\widehat{L}_0(\theta^{(s)}), \gamma, \tau, q, T_s)$ and send it to each node.

9          Each node computes $\nabla\widehat{L}_k(\theta^{(s)})$ in parallel and send it to the center.

10        Set $\bar{\nabla}_k\widehat{F}(w^{(s)}) = -\nabla\widehat{L}_k(\theta^{(s)})^\top h^{(s)}$ for $k = 1, \ldots, K$

11      $w^{(s+1)} = \arg\min_{w \in \Delta_K^b} \left\langle \bar{\nabla}\widehat{F}(w^{(s)}), w \right\rangle + \frac{1}{2\eta}\|w - w^{(s)}\|^2$

12    **end**

13    **Return**: $w^{(\bar{s})}$ with $\bar{s}$ sampled randomly from $\{0, 1, \ldots, S-1\}$.

---

The convergence result of Algorithm 3 can be proved in a standard way (e.g., see Theorem 2.1 in Ghadimi & Wang (2018)). We present it below only for the sake of completeness.

**Theorem 4.** *Suppose Assumption 1 holds. Let $R := \max_{w \in \Delta_K^b} \|\widehat{\theta}(w)\|$ and $\ell_F$ and $\gamma_0$ defined as in (12) and (17). Suppose $\eta = \frac{1}{3\ell_F}$ in Algorithm 3. There exist constants $A_1$, $A_2$ and $A_3$ that only depend on $\ell_0$, $\ell_1$, $\ell_2$, $\mu$, $R$, $q$ and $K$ but not on $\tau$ such that the following statements hold.*

- *Suppose $\tau = 1$, $\gamma = \gamma_0$ and $T_s = \frac{1}{\gamma_0\mu} \ln\left(\frac{A_1(s+1)^2}{\gamma_0}\right)$. Algorithm 3 finds an $\epsilon$-stationary solution of ($\widehat{P}$) with $\tilde{O}\left(\epsilon^{-2}\right)$ rounds of communication.*

- *Suppose $\tau > 1$, $\gamma = \frac{1}{M_s'}$ and $T_s = \mu^{-1}M_s'\ln\left(M_s'^3\right)$, where*

$$M_s' = \max\left\{\frac{1}{\gamma_0}, (s+1)\sqrt{[A_1 + A_2(\tau-1) + A_3(\tau-1)^2]}\right\}, s = 0, 1, \ldots. \qquad (51)$$

*Algorithm 3 finds an $\epsilon$-stationary solution of ($\widehat{P}$) with $\tilde{O}\left(\epsilon^{-4}\right)$ rounds of communication.*

*Proof.* Let $E_s := \left\|\nabla\widehat{F}(w^{(s)}) - \bar{\nabla}\widehat{F}(w^{(s)})\right\|$. Since $w^{(s+1)} = \text{Proj}_{\Delta_K^b}(w^{(s)} - \eta\bar{\nabla}\widehat{F}(w^{(s)}))$, by the property of projection mapping, we have

$$\left\|w^{(s)} - \text{Proj}_{\Delta_K^b}(w^{(s)} - \eta\nabla\widehat{F}(w^{(s)}))\right\|^2$$
$$\leq \; 2\left\|w^{(s)} - w^{(s+1)}\right\|^2 + 2\left\|\text{Proj}_{\Delta_K^b}(w^{(s)} - \eta\bar{\nabla}\widehat{F}(w^{(s)})) - \text{Proj}_{\Delta_K^b}(w^{(s)} - \eta\nabla\widehat{F}(w^{(s)}))\right\|^2$$
$$\leq \; 2\left\|w^{(s)} - w^{(s+1)}\right\|^2 + 2\eta^2 E_s^2. \qquad (52)$$

By the definition of $w^{(s+1)}$ and the $\frac{1}{\eta}$-strong convexity of function $\left\langle \bar{\nabla}\widehat{F}(w^{(s)}), w \right\rangle + \frac{1}{2\eta}\|w - w^{(s)}\|^2$, we have, for any $w \in \Delta_K^b$

$$\left\langle \bar{\nabla}\widehat{F}(w^{(s)}), w^{(s+1)} - w^{(s)} \right\rangle + \frac{1}{2\eta}\|w^{(s+1)} - w^{(s)}\|^2 + \frac{1}{2\eta}\|w - w^{(s+1)}\|^2$$
$$\leq \; \left\langle \bar{\nabla}\widehat{F}(w^{(s)}), w - w^{(s)} \right\rangle + \frac{1}{2\eta}\|w - w^{(s)}\|^2. \qquad (53)$$

Taking $w = w^{(s)}$ in (53) gives

$$\left\langle \bar{\nabla}\widehat{F}(w^{(s)}), w^{(s+1)} - w^{(s)}\right\rangle + \frac{1}{\eta}\|w^{(s+1)} - w^{(s)}\|^2 \leq 0. \tag{54}$$

Since $\widehat{F}$ is $\ell_F$-Lipschitz continuous and $\eta \leq \frac{1}{\ell_F}$, we have

$$\widehat{F}(w^{(s+1)}) - \widehat{F}(w^{(s)}) \leq \left\langle \nabla\widehat{F}(w^{(s)}), w^{(s+1)} - w^{(s)}\right\rangle + \frac{1}{2\eta}\|w^{(s+1)} - w^{(s)}\|^2. \tag{55}$$

Adding (54) and (55) gives us

$$\widehat{F}(w^{(s+1)}) - \widehat{F}(w^{(s)}) + \frac{1}{2\eta}\|w^{(s+1)} - w^{(s)}\|^2 \leq \left\langle \nabla\widehat{F}(w^{(s)}) - \bar{\nabla}\widehat{F}(w^{(s)}), w^{(s+1)} - w^{(s)}\right\rangle$$

$$\leq E_s\|w^{(s+1)} - w^{(s)}\| \leq 2E_s,$$

which, together with (52), implies

$$\frac{1}{4\eta}\left\|w^{(s)} - \mathrm{Proj}_{\Delta_K^b}(w^{(s)})\right\|^2 \leq \widehat{F}(w^{(s)}) - \widehat{F}(w^{(s+1)}) + 2E_s + \frac{\eta E_s^2}{2}.$$

Summing this inequality and taking expectation give us

$$\eta^{-2}\mathbb{E}\left[\left\|w^{(\bar{s})} - \mathrm{Proj}_{\Delta_K^b}(w^{(\bar{s})})\right\|^2\right] \leq \frac{4}{\eta S}\left[\widehat{F}(w^{(0)}) - \widehat{F}(\widehat{w})\right] + \frac{8}{\eta S}\sum_{s=0}^{S-1}\mathbb{E}[E_s] + \frac{2}{S}\sum_{s=0}^{S-1}\mathbb{E}[E_s^2].$$

When $\mathbb{E}[E_s^2] \leq \frac{1}{(s+1)^2}$, the inequality above implies

$$\eta^{-2}\mathbb{E}\left\|w^{(\bar{s})} - \mathrm{Proj}_{\Delta_K^b}(w^{(\bar{s})})\right\|^2 \leq \frac{4}{\eta S}\left[\widehat{F}(w^{(0)}) - \widehat{F}(\widehat{w})\right] + \frac{8\log(S)}{\eta S} + \frac{\pi^2}{3S}.$$

This means, as long as $\mathbb{E}[E_s^2] \leq \frac{1}{(s+1)^2}$, Algorithm 3 finds an $\epsilon$-stationary solution of $(\widehat{P})$ in $\tilde{O}(\epsilon^{-2})$ iterations.

Note that Lemma 4 still holds with $w_{\mathrm{md}}^{(s)}$ replaced by $w^{(s)}$ in Algorithm 3. Let $A_1 = C_1 C_2$, $A_2 = C_1 C_3$ and $A_3 = C_1 C_4$ with $C_1, C_2, C_3$ and $C_4$ defined in Lemma 4.

Suppose $\tau = 1$, $\gamma = \gamma_0$ and $T_s = \frac{1}{\gamma_0 \mu}\ln\left(\frac{A_1(s+1)^2}{\gamma_0}\right)$. By Lemma 4, we have $\mathbb{E}[E_s^2] \leq \frac{1}{(s+2)^2}$. In iteration $s$ of Algorithm 3, the total number of rounds of communication needed in Local-SVRG is $O(T_s) = O(\ln(s))$ so that means the total number of rounds is $\tilde{O}(\epsilon^{-2})$.

Suppose $\tau > 1$, $\gamma = \frac{1}{M_s'}$ and $T_s = \mu^{-1} M_s' \ln\left(M_s'^3\right)$. By Lemma 4, we have $\mathbb{E}[E_s^2] \leq \frac{1}{(s+2)^4}$. In iteration $s$ of Algorithm 3, the total number of rounds of communication needed in Local-SVRG is $O(T_s/\tau) = O(s)$ so that means the total number of rounds is $\tilde{O}(\epsilon^{-4})$. □

**Remark 1.** *The federated bilevel optimization methods by Li et al. (2022) and Tarzanagh et al. (2022) can find an $\epsilon$-stationary point within $\tilde{O}(\epsilon^{-3})$ and $\tilde{O}(\epsilon^{-4})$ rounds of communication, respectively. In the first setting of Theorem 4 ($\tau = 1$), Algorithm 3 has complexity of $\tilde{O}(\epsilon^{-2})$, which is better than Li et al. (2022) and Tarzanagh et al. (2022). We want to point out that the lower complexity of Algorithm 3 is because it utilizes the finite-sum structure in $(\widehat{P})$, which allows computing a deterministic gradient infrequently to accelerate the convergence. However, Li et al. (2022) and Tarzanagh et al. (2022) both consider objective functions given in expectation, which does not allow computing a deterministic gradient in general. One can check the following Table 2 for a comparison.*

## G  ADDITIONAL MATERIALS FOR NUMERICAL EXPERIMENTS

In this section, we present additional details and results of our numerical experiments in Section 5. The CNN we train in the experiments consists of two layers of 2D convolution, each equipped with 2D batch normalization and ReLU activation, and followed by a fully connected layer to generate

Table 2: Comparison in **Remark 1**.

| Paper | Structure | Case | Number of Rounds |
|---|---|---|---|
| This work | Finite-sum | Convex | $\tilde{O}(\epsilon^{-0.5})$ |
| This work | Finite-sum | Non-convex | $\tilde{O}(\epsilon^{-2})$ |
| Chen et al. (2022) | Deterministic | Non-convex | $\tilde{O}(\epsilon^{-2})$ |
| Li et al. (2022) | Expectation | Non-convex | $\tilde{O}(\epsilon^{-3})$ |
| Chen et al. (2022) | Expectation | Non-convex | $\tilde{O}(\epsilon^{-4})$ |
| Tarzanagh et al. (2022) | Expectation | Non-convex | $\tilde{O}(\epsilon^{-4})$ |

predictions. The first convolution layer is set to output the same number of channels as the input, and uses kernels with a size of 4, a stride of 4 and one padding. The second convolution layer returns two output channels for Fashion-MNIST and MNIST and five for CIFAR-10 and ImageNet, and uses kernels with a size of 2, a stride of 2 and one padding. All experiments are conducted with PyTorch 1.9.0 and CUDA 11.1 computing platform on a computer with the CPU Intel Xeon Gold 6330@2.0GHz (Turbo up to 3.1GHz) and the GPU NVIDIA GeForce RTX 2080 Ti.

## G.1 DATA GENERATION WITH DIFFERENT CLASS DISTRIBUTIONS (SETTING 1)

In this section, we describe in details how we generate $D^{\text{valid}}$, $D^{\text{train}}$ and $D^{\text{test}}$ from each original dataset for our experiments under Setting 1.

### G.1.1 FASHION-MNIST

Fashion-MNIST (Xiao et al., 2017) contains a training set of 60,000 images and a testing set of 10,000 images. Each image is in grayscale, has a size of $28 \times 28$, and is associated with a label from ten classes: 0: T-shirt/top, 1: Trouser, 2: Pullover, 3: Dress, 4: Coat, 5: Sandal, 6: Shirt, 7: Sneaker, 8: Bag and 9: Ankle boot. We merge the ten classes into four classes as follows:

C1: Classes 2, 4 and 6 which include long-sleeve upper-body clothes;

C2: Classes 0 and 3 which include short-sleeve upper-body clothes;

C3: Classes 1 and 8 which include pants and bags;

C4: Classes 5, 7 and 9 which include only shoes.

Note that these four merged classes are only used for generating data. In the classification task, we still have ten classes. This is the same for the other three datasets. We set $n_{\text{train}} = 4000$, $n_{\text{valid}} = 500$ and $n_{\text{test}} = 5000$. Each image is sampled from one of the four merged classes with a probability distribution $(P_1, P_2, P_3, P_4)$. Once a merged class is chosen, each image in that merged class has an equal chance to be sampled. For $D_k^{\text{train}}$ with $k \in \mathcal{J}$, we sample data from the training set with $P_1 = 0.42$, $P_2 = 0.08$, $P_3 = 0.38$ and $P_4 = 0.12$. For $D_k^{\text{train}}$ for $k \in \mathcal{J}_M$, we sample data with $P_1 = 0.12$, $P_2 = 0.38$, $P_3 = 0.08$ and $P_4 = 0.42$. Depending on $p_0$ is the distribution of $\mathcal{J}_m$ or $\mathcal{J}_M$, $D^{\text{valid}}$ and $D^{\text{test}}$ are sampled from $p_0$ the corresponding distribution. Note that $D^{\text{valid}}$ and $D^{\text{test}}$ are sampled from the training set and testing set of the original data, respectively, although they have the same probability distribution over the four merged classes. Since $D^{\text{valid}}$ and $D^{\text{test}}$ are generated in the similar way for the other three datasets, we will only discuss the generation of $D_k^{\text{train}}$'s in the subsequent sections.

### G.1.2 MNIST

MNIST (Deng, 2012) contains a training set of 60,000 images and a testing set of 10,000 images. Each image is a handwritten digit in grayscale and has a size of $28 \times 28$. Since MNIST has the same number of classes, same class distribution, and same data size as Fashion-MNIST, we directly apply the same procedure in Section G.1.1 to sample data. In particular, we merge the ten digits into four classes as follows:

C1: Digits 2, 4 and 6;

C2: Digits 0 and 3;

C3: Digits 1 and 8;

C4: Digits 5, 7 and 9.

We set $n_{\text{train}} = 4000$, $n_{\text{valid}} = 500$ and $n_{\text{test}} = 5000$. Following Section G.1.1, for $D_k^{\text{train}}$ with $k \in \mathcal{J}_m$, we sample data from the four merged classes with $P_1 = 0.42$, $P_2 = 0.08$, $P_3 = 0.38$ and $P_4 = 0.12$. For $D_k^{\text{train}}$ with $k \in \mathcal{J}_M$, we sample data with $P_1 = 0.12$, $P_2 = 0.38$, $P_3 = 0.08$ and $P_4 = 0.42$.

### G.1.3 CIFAR-10

CIFAR-10 (Krizhevsky et al., 2009) contains a training set of 50,000 images and a testing set of 10,000 images. Each image is in color, has a size of $32 \times 32$, and is associated with a label from ten classes: 0: airplane, 1: automobile, 2: bird, 3: cat, 4: deer, 5: dog, 6: frog, 7: horse, 8: ship and 9: truck. We merge the ten classes into four classes as follows:

C1: Classes 1 and 9 which are related to ground transportation;

C2: Classes 0 and 8 which are related to non-ground transportation;

C3: Classes 2, 3 and 4 which form a set of animals;

C4: Classes 5, 6 and 7 which form another set of animals.

We set $n_{\text{train}} = 4000$, $n_{\text{valid}} = 500$ and $n_{\text{test}} = 5000$. Similar to the procedure with Fashion-MNIST, for $D_k^{\text{train}}$ with $k \in \mathcal{J}_m$, we sample data from the four merged classes with $P_1 = 0.36$, $P_2 = 0.04$, $P_3 = 0.54$ and $P_4 = 0.06$. For $D_k^{\text{train}}$ with $k \in \mathcal{J}_M$, we sample data with $P_1 = 0.04$, $P_2 = 0.36$, $P_3 = 0.06$ and $P_4 = 0.54$.

### G.1.4 DOWNSAMPLED IMAGENET

Downsampled ImageNet (Chrabaszcz et al., 2017) is created by downsampling each image in ImageNet (Deng et al., 2009) to $32 \times 32$ pixels without changing the class labels. Just as ImageNet, downsampled ImageNet has 1000 classes and we choose ten classes and merge them into four classes as follows. (The class labels listed below are consistent with ImageNet.)

C1: Classes $7, 9, 10, 29, 54, 75, 84$ and $189$, which are cats or animals similar to cat;

C2: Classes $61, 66, 68, 101, 114, 124, 131$ and $148$, which are dogs or animals similar to cat;

C2: Classes $383, 397, 403, 404, 405, 406, 412, 414, 420, 426, 433$ and $434$, which are all birds;

C3: Classes $224, 441, 442, 443, 444, 445, 449, 453, 454, 498, 499$ and $500$, which are either fishes or frogs.

We set $n_{\text{train}} = 4000$, $n_{\text{valid}} = 1500$, $n_{\text{test}} = 1000$. For $D_k^{\text{train}}$ for $k \in \mathcal{J}_m$, we sample data from the four merged classes with $P_1 = 0.36$, $P_2 = 0.04$, $P_3 = 0.54$ and $P_4 = 0.06$. For $D_k^{\text{train}}$ with $k \in \mathcal{J}_M$, we sample data with $P_1 = 0.04$, $P_2 = 0.36$, $P_3 = 0.06$ and $P_4 = 0.54$.

### G.1.5 COVTYPE

Covtype (Blackard & Dean, 1999) is an imbalanced dataset of 581,012 instances. Each instance records ten features in integer value and two categorical features one-hot encoded into 44 binary values, and is associated with a label from seven classes: 1: Spruce/Fir, 2: Lodgepole Pine, 3: Ponderosa Pine, 4: Cottonwood/Willow, 5: Aspen, 6: Douglas-fir, 7: Krummholz. According to the imbalance on class labels, we merge the seven classes into four classes as follows:

C1: Classes 1 which consists of approximately $36.5\%$ of the raw data;

C2: Classes 2 which consists of approximately $48.8\%$ of the raw data;

C3: Classes 3 which consists of approximately $6.2\%$ of the raw data;

C4: Classes 4, 5, 6 and 7 which consist of the rest.

We set $n_{\text{train}} = 40000$, $n_{\text{valid}} = 5000$ and $n_{\text{test}} = 50000$. Similar to the procedure with Fashion-MNIST, for $D_k^{\text{train}}$ with $k \in \mathcal{J}_m$, we sample data from the four merged classes with $P_1 = 0.72$, $P_2 = 0.14$, $P_3 = 0.12$ and $P_4 = 0.02$. For $D_k^{\text{train}}$ with $k \in \mathcal{J}_M$, we sample data with $P_1 = 0.10$, $P_2 = 0.82$, $P_3 = 0.06$ and $P_4 = 0.02$.

We train a liner multi-class logistic regression model using the same computing platform in this experiment as before. We use a mini batch of size $400$ to construct the stochastic gradients in all methods. In the Bi-level method, Algorithm 3 is applied to $(\widehat{P})$ with $b = 1/3$ and five epochs are performed within each call of Local-SVRG (i.e., $T_s = 5n_{\text{train}}/400$). We set $q = 1/100$. Other parameters are the same as our choice in Section 5. We only run Setting 1 with one random seed on this dataset. The results are shown in Figure 3.

## G.2 Data Generation with Different Class Distributions and Label Permutation (Setting 2)

In this section, we discuss in details how $D^{\text{train}}$, $D^{\text{valid}}$ and $D^{\text{test}}$ are generated from each original dataset for our experiments under Setting 2. For each dataset, we first sample data $D^{\text{train}}$, $D^{\text{valid}}$ and $D^{\text{test}}$ in the same way as Setting 1 described in Section G.1. Then we permute the class labels in $D_k^{\text{train}}$ with $k \in \mathcal{J}_M$. For each dataset, the class labels in $D_k^{\text{train}}$ with $k \in \mathcal{J}_m$ are unchanged. If $p_0$ is the distribution of $\mathcal{J}_M$, the labels of $D^{\text{valid}}$ and $D^{\text{test}}$ are permuted in the same way as $D_k^{\text{train}}$ with $k \in \mathcal{J}_M$. If $p_0$ is the distribution of $\mathcal{J}_m$, the labels of $D^{\text{valid}}$ and $D^{\text{test}}$ are unchanged.

We then describe how the class labels in $D_k^{\text{train}}$ for $k \in \mathcal{J}_M$ are permuted for each dataset. For Fashion-MNIST and MNIST, we permute the class labels in $D_k^{\text{train}}$ for $k \in \mathcal{J}_M$ by changing label 2 to 0, 0 to 1, 1 to 5, and 5 to 2. For CIFAR-10, we permute the class labels in $D_k^{\text{train}}$ for $k \in \mathcal{J}_M$ by changing label 1 to 0, 0 to 2, 2 to 5, and 5 to 1. For downsampled ImageNet, we permute the class labels in $D_k^{\text{train}}$ for $k \in \mathcal{J}_M$ by changing label 7 to 61, 61 to 383, 383 to 224, 224 to 9, 9 to 66, 66 to 397, 397 to 441, 441 to 10, 10 to 68, 68 to 403, 403 to 442, and 442 to 7.

## G.3 Data Generation with Different Class Distributions, Label Permutation and/or Random Rotation (Settings 3 and 4)

In this section, we discuss in details how we generate $D^{\text{valid}}$, $D^{\text{train}}$ and $D^{\text{test}}$ from each original dataset for our experiments under Settings 3 and 4.

Under Setting 3, we first generate data in the same way as in Setting 1 described in Section G.1. Then, we randomly choose a rotation direction, clockwise or anti-clockwise, and rotate each image in $D_k^{\text{train}}$ with $k \in \mathcal{J}_M$ toward that direction for 90 degrees. Under Setting 4, we first generate data in the same way as in Setting 2 described in Section G.2. Then, we apply the same rotation procedure as we do in Setting 3.

## G.4 Additional Numerical Results

In this section, we first plot how the weight $w_k$ for each node evolves during the Bi-level method under Setting 2, 3 and 4 respectively in Figure 4, Figure 5 and Figure 6. We show the results when $p_0$ is the distribution of the majority and the minority groups separately.

Since the performance of an algorithm may fluctuate during training, we are interested in comparing the methods in the best performance they achieved during training. To do so, we save the model generated by each method at the iteration where the highest accuracy is achieved on the validation data. Then, we report the performance of the saved model's by each method on the testing set in Table 3. Again, we show the results when $p_0$ is the distribution of the majority and the minority groups separately. Our method outperforms the baselines in most of the cases.

Next, we plot the test (top-1) accuracy each method obtains during iterations for the minority group and the majority group in Figure 8 and Figure 9, respectively, where the horizontal axis represents the cumulative number of data points each method processes in parallel.

At last, we give the results of the ablation study. We choose the case of Setting 2 Seed 1, and consider four choices on the size of the validation dataset $D^{\text{valid}}$, i.e. $n_0$. The choices are 100, 200,

500 and 1000 for Fashion-MNIST, MNIST and CIFAR- 10, and 1000, 1200, 1500 and 2000 for the downsampled $32 \times 32$ ImageNet. Similarly, we save the model generated by our Bi-level method at the iteration where the highest accuracy is achieved on the respective validation data. Then, we report the test (top-1) accuracy each method obtains during iterations for the minority group in Table 4 and the majority group in Table 5 .

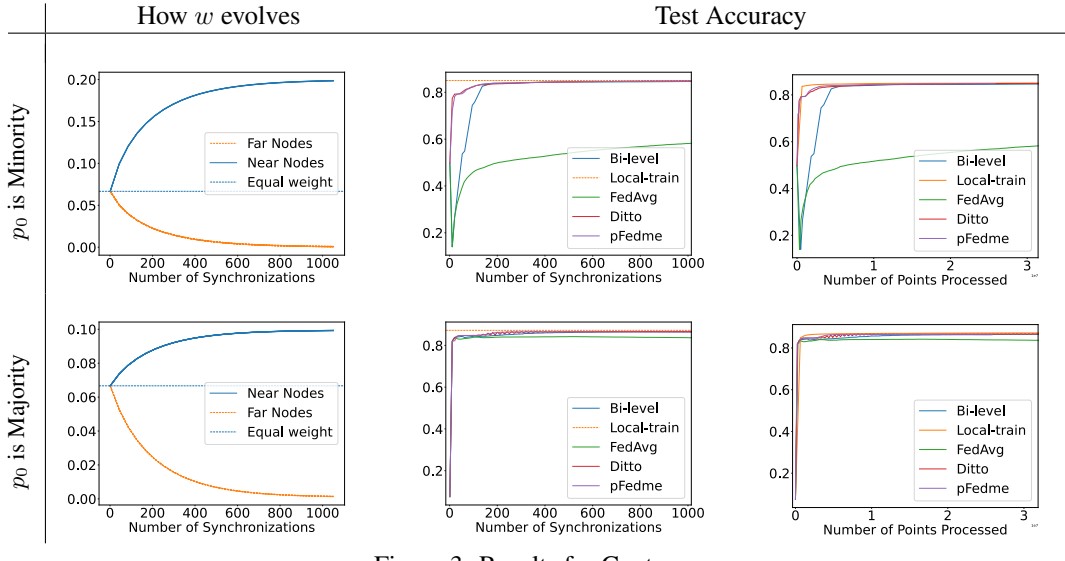

Figure 3: Results for Covtype.

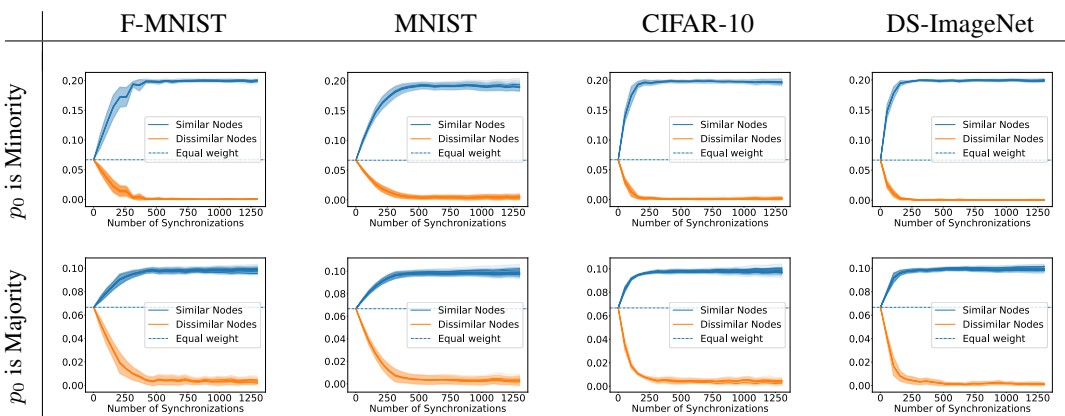

Figure 4: How $w$ evolves during the Bi-level method under Setting 2.

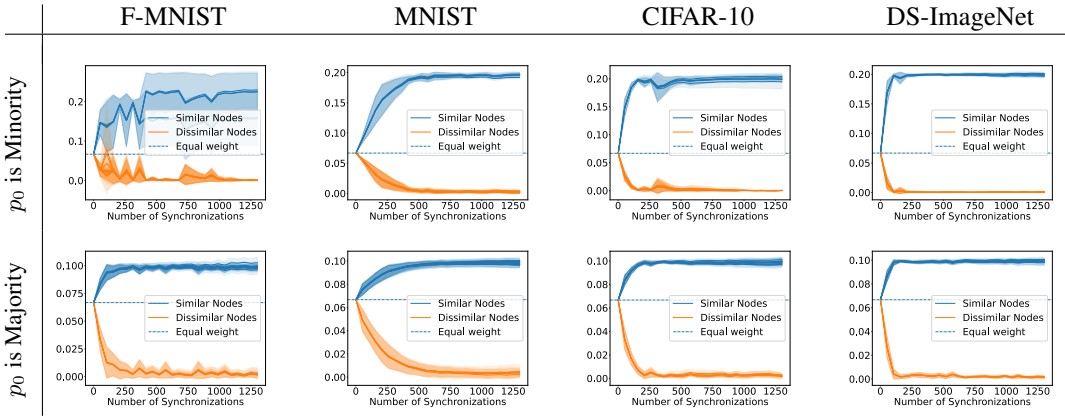

Figure 5: How $w$ evolves during the Bi-level method under Setting 3.

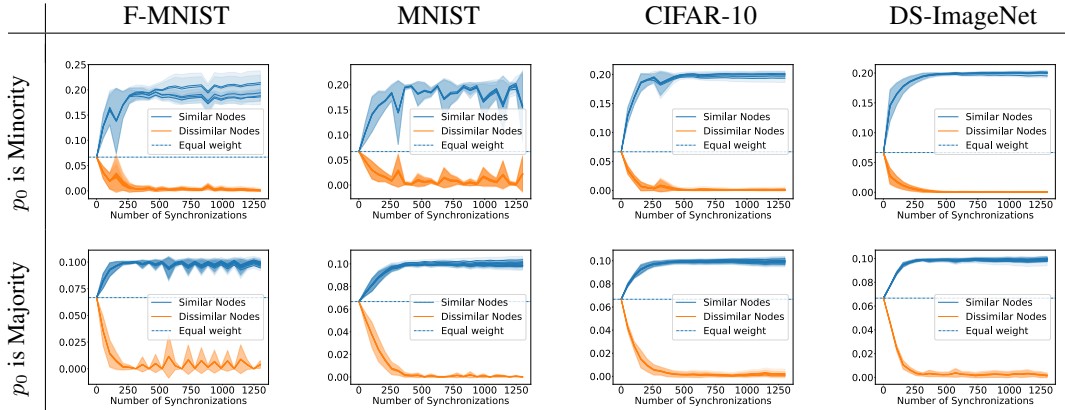

| | F-MNIST | MNIST | CIFAR-10 | DS-ImageNet |

Figure 6: How $w$ evolves during the Bi-level method under Setting 4.

Table 3: Test accuracy when reaching the highest validation accuracy.

| Setting 1 | | Minority Model | | | | Majority Model | | |
|---|---|---|---|---|---|---|---|---|
| Method\Data | F-MNIST | MNIST | CIFAR-10 | DS-ImageNet | F-MNIST | MNIST | CIFAR-10 | DS-ImageNet |
| Bi-level | **0.7758±0.0059** | **0.8824±0.0198** | **0.5175±0.0065** | **0.2668±0.0079** | **0.8364±0.0121** | 0.8443±0.0096 | **0.5928±0.0068** | **0.2630±0.0091** |
| Local-train | 0.6926±0.0175 | 0.7850±0.0269 | 0.3036±0.0102 | 0.1108±0.0127 | 0.7427±0.0110 | 0.7550±0.0095 | 0.3568±0.0144 | 0.1150±0.0072 |
| FedAvg | 0.7507±0.0097 | 0.8297±0.0171 | 0.2965±0.0075 | 0.2008±0.0117 | 0.8327±0.0119 | **0.8484±0.0123** | 0.5705±0.0052 | 0.2612±0.0116 |
| Ditto | 0.7297±0.0146 | 0.8358±0.0201 | 0.4361±0.0166 | 0.1582±0.0037 | 0.8086±0.0141 | 0.8059±0.0053 | 0.5021±0.0204 | 0.1874±0.0126 |
| pFedMe | 0.7418±0.0235 | 0.8568±0.0204 | 0.4860±0.0067 | 0.2084±0.0068 | 0.8297±0.0070 | 0.8320±0.0162 | 0.5588±0.0110 | 0.2324±0.0087 |
| Setting 2 | | Minority Model | | | | Majority Model | | |
| Method\Data | F-MNIST | MNIST | CIFAR-10 | DS-ImageNet | F-MNIST | MNIST | CIFAR-10 | DS-ImageNet |
| Bi-level | **0.7754±0.0082** | **0.8580±0.0283** | **0.5148±0.0051** | **0.2734±0.0052** | **0.8332±0.0063** | **0.8384±0.0196** | **0.5979±0.0044** | **0.2694±0.0104** |
| Local-train | 0.6763±0.0116 | 0.7874±0.0129 | 0.3159±0.0110 | 0.1260±0.0033 | 0.7343±0.0104 | 0.7650±0.0146 | 0.3594±0.0086 | 0.1276±0.0110 |
| FedAvg | 0.5686±0.0211 | 0.5809±0.0405 | 0.2870±0.0106 | 0.1676±0.0044 | 0.7824±0.0062 | 0.7746±0.0121 | 0.5431±0.0087 | 0.2542±0.0089 |
| Ditto | 0.7204±0.0059 | 0.7841±0.0217 | 0.3951±0.0121 | 0.1536±0.0178 | 0.7889±0.0093 | 0.7841±0.0089 | 0.4946±0.0131 | 0.1724±0.0068 |
| pFedMe | 0.7207±0.0130 | 0.7506±0.0224 | 0.4422±0.0131 | 0.1936±0.0113 | 0.8091±0.0044 | 0.7911±0.0283 | 0.5574±0.0104 | 0.2146±0.0081 |
| Setting 3 | | Minority Model | | | | Majority Model | | |
| Method\Data | F-MNIST | MNIST | CIFAR-10 | DS-ImageNet | F-MNIST | MNIST | CIFAR-10 | DS-ImageNet |
| Bi-level | **0.7726±0.0170** | **0.8736±0.0055** | **0.5235±0.0060** | **0.2680±0.0072** | **0.8234±0.0119** | **0.8342±0.0135** | **0.5861±0.0064** | **0.2788±0.0142** |
| Local-train | 0.6762±0.0126 | 0.7803±0.0204 | 0.3061±0.0107 | 0.1162±0.0054 | 0.7262±0.0074 | 0.7698±0.0100 | 0.3632±0.0083 | 0.1224±0.0071 |
| FedAvg | 0.5940±0.0211 | 0.6225±0.0195 | 0.3084±0.0099 | 0.1756±0.0088 | 0.7702±0.0143 | 0.7632±0.0171 | 0.5705±0.0123 | 0.2676±0.0074 |
| Ditto | 0.6856±0.0247 | 0.7452±0.0160 | 0.4184±0.0149 | 0.1626±0.0093 | 0.7628±0.0115 | 0.7742±0.0167 | 0.4986±0.0175 | 0.1792±0.0133 |
| pFedMe | 0.7123±0.0239 | 0.7203±0.0201 | 0.4535±0.0068 | 0.1938±0.0144 | 0.7871±0.0104 | 0.7650±0.0341 | 0.5563±0.0038 | 0.2188±0.0122 |
| Setting 4 | | Minority Model | | | | Majority Model | | |
| Method\Data | F-MNIST | MNIST | CIFAR-10 | DS-ImageNet | F-MNIST | MNIST | CIFAR-10 | DS-ImageNet |
| Bi-level | **0.7658±0.0117** | **0.8588±0.0133** | **0.5172±0.0107** | **0.2698±0.0115** | **0.8160±0.0051** | **0.8455±0.0064** | **0.5935±0.0074** | **0.2678±0.0062** |
| Local-train | 0.6809±0.0079 | 0.7645±0.0270 | 0.3086±0.0052 | 0.1148±0.0059 | 0.7012±0.0066 | 0.7476±0.0282 | 0.3709±0.0079 | 0.1172±0.0087 |
| FedAvg | 0.5648±0.0481 | 0.5475±0.0205 | 0.3612±0.0181 | 0.1722±0.0090 | 0.7644±0.0258 | 0.7577±0.0127 | 0.5529±0.0077 | 0.2600±0.0068 |
| Ditto | 0.6974±0.0130 | 0.7176±0.0185 | 0.3970±0.0105 | 0.1538±0.0077 | 0.7646±0.0085 | 0.7608±0.0063 | 0.4986±0.0102 | 0.1744±0.0159 |
| pFedMe | 0.7035±0.0187 | 0.6932±0.0140 | 0.4548±0.0069 | 0.1794±0.0114 | 0.7910±0.0110 | 0.7688±0.0171 | 0.5489±0.0060 | 0.2232±0.0063 |

Table 4: Test accuracy's of the minority model by the Bi-level method using validation sets of different sizes.

| | $n_{\text{valid}}$ | F-MNIST | MNIST | CIFAR-10 | $n_{\text{valid}}$ | DS-ImageNet |
|---|---|---|---|---|---|---|
| | 100 | 0.7782 | 0.8444 | 0.5226 | 1000 | 0.2850 |
| | 200 | 0.7830 | 0.8778 | 0.5170 | 1200 | 0.2700 |
| Test Accuracy | 500 | 0.7756 | 0.8108 | 0.5100 | 1500 | 0.2740 |
| | 1000 | **0.7842** | **0.8852** | **0.5292** | 2000 | **0.2930** |

Table 5: Test accuracy's of the majority model by the Bi-level method using validation sets of different sizes.

| | $n_{\text{valid}}$ | F-MNIST | MNIST | CIFAR-10 | $n_{\text{valid}}$ | DS-ImageNet |
|---|---|---|---|---|---|---|
| | 100 | 0.8286 | 0.7946 | 0.5664 | 1000 | 0.2600 |
| | 200 | 0.8362 | 0.8480 | 0.5620 | 1200 | 0.2710 |
| Test Accuracy | 500 | 0.8216 | 0.8044 | **0.5920** | 1500 | **0.2920** |
| | 1000 | **0.8394** | **0.8514** | 0.5890 | 2000 | 0.2640 |

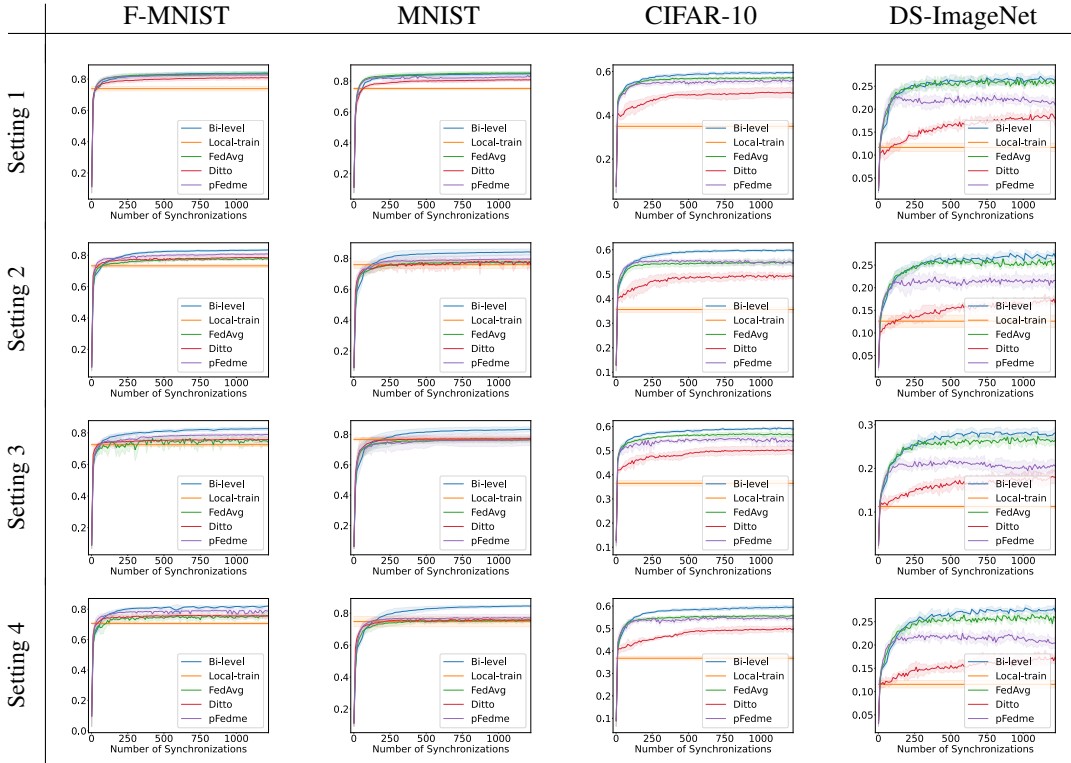

Figure 7: Comparison in test accuracy for the majority group vs number of synchronizations.

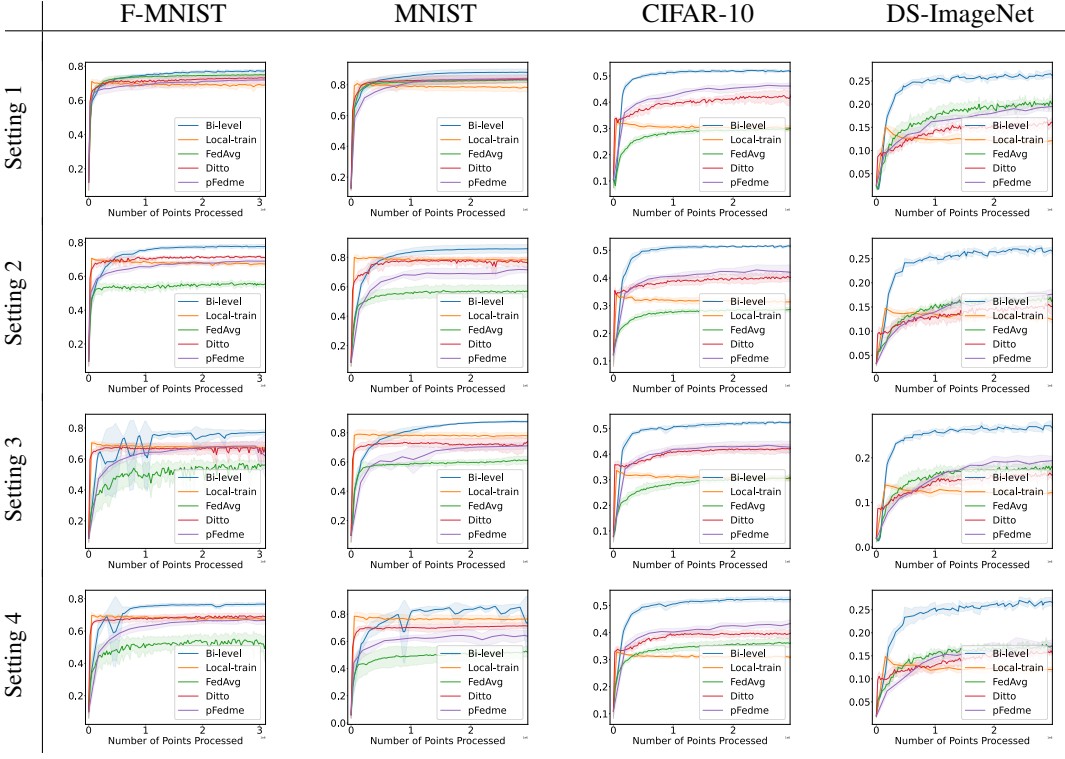

Figure 8: Comparison in test accuracy for the minority group vs number of points processed.

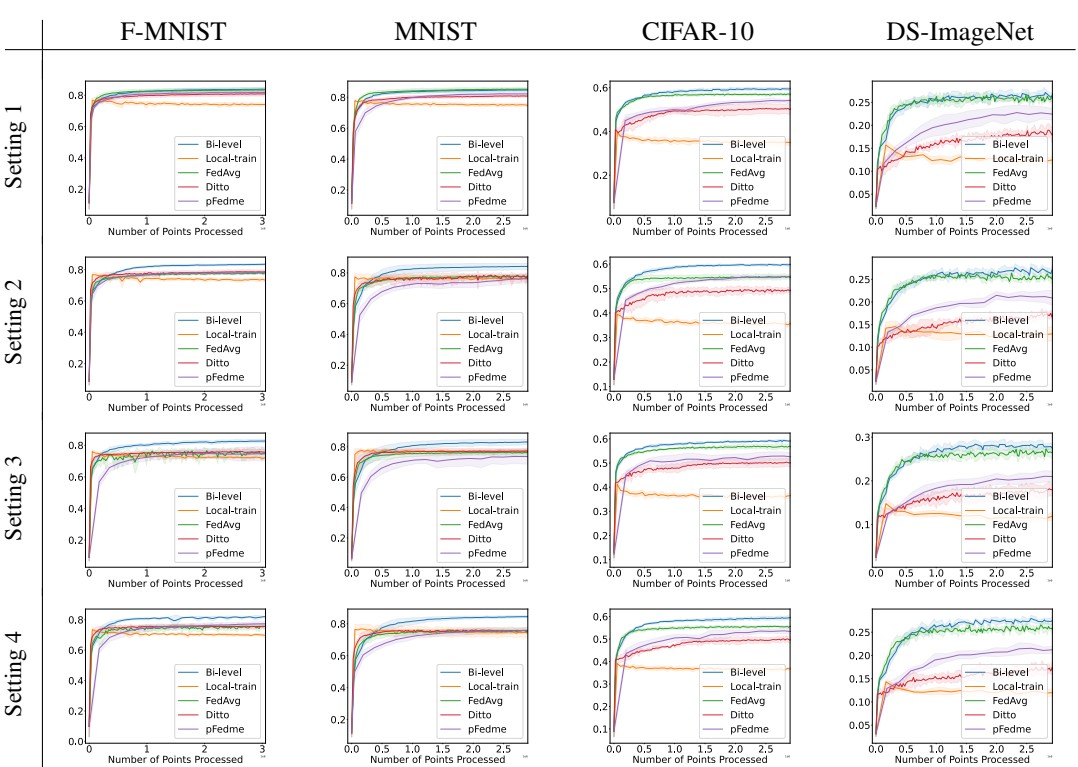

Figure 9: Comparison in test accuracy for the majority group vs number of points processed.

