# OpenReview forum: "Federated Learning on Adaptively Weighted Nodes by Bilevel Optimization"
_ICLR.cc/2023/Conference — Submitted to ICLR 2023_

### Official Review · Reviewer_9pHr · 2022-10-19

**Confidence:** 3
**Correctness:** 3
**Technical Novelty And Significance:** 3
**Empirical Novelty And Significance:** 3
**Recommendation:** 6

**Clarity, Quality, Novelty And Reproducibility:**

This paper has good quality, clarity, and contributions in both theoretical and empirical aspects.

**Strength And Weaknesses:**

Strengths:
- The proposed method is interesting.
- The theoretical analysis provides additional guarantees.
- The results show the effectiveness of the proposed method.

Weaknesses:
- The bilevel optimization highly depends on the size, quality, and distribution of the validation dataset. In many real-world scenarios, high-quality validation data is not available on local devices (e.g., personal computers or smartphones).
- Since this paper aims to bolster local model performance, it is required to compare several more baseline methods for personalized federated learning.
- The experiments are conducted only on image classification datasets. I suggest the authors consider using some recommendation datasets (e.g., Movielens) since the user data in these datasets are naturally heterogeneous (and more realistic than simulated data).
- The real communication costs of the proposed method and existing methods are not reported. It is suggested to include the results to confirm that the proposed method is "communication-efficient".

**Summary Of The Paper:**

This paper presents a bilevel optimization-based federated learning method that can assign different nodes different weights to optimize the performance on a validation dataset. The authors present thorough theoretical analysis of the problem and verify that learning dynamic weights is better than using evenly distributed weights. Experimental analysis also shows the superiority of the proposed method over several baseline methods.

**Summary Of The Review:**

This paper has good contributions and the paper is clearly written, but the experiments are not very solid and certain claims are not fully verified. Thus, my recommendation is a weak accept.

---

> ### Author Response · Authors · 2022-11-18
> **Response to Reviewer 9pHr**
>
> Thank you so much for your time and comments.  The following are responses to your questions and concerns.
>
> **Q: The bilevel optimization highly depends on the size, quality, and distribution of the validation dataset. In many real-world scenarios, high-quality validation data is not available on local devices (e.g., personal computers or smartphones).**
>
> A: We agree with the reviewer that high-quality validation data is not always available, e.g., on smartphones. We actually focus on the cross-silo scenario instead of the cross-device scenario. In the cross-silo scenario, the center and nodes we consider are institutes such as hospitals. In this case, the validation data is generally available although the size varies with institutes. This is related to a question raised by Reviewer F6ZM, who wondered if the size of validation data affects the performance. To answer this question, we choose Setting 2 in our experiments and conduct the experiments using validation sets of different sizes and watch how the performances are affected. For FashionMNIST, MNIST and Cifar-10, we run experiments with sizes 100, 200, 500 and 1000. For ImageNet, we run experiments with sizes 1000, 1200, 1500 and 2000. Please see Section F.4, or more specifically, see Table 3 and Table 4 on page 34 in the revision we submitted. Overall, the results show that a larger validation set improve the testing performance in general. As long as the size of validation set is not too small (e.g. only 100), the testing performance of our method is stable with an average change less than 5\% when the size of validation set changes.
>
> **Q: Since this paper aims to bolster local model performance, it is required to compare several more baseline methods for personalized federated learning.**
>
> A: In Section 5, we have included two personalized federated learning methods, Ditto and pFedMe. One empirical finding is that, if the personalized method starts with a model that is trained without adaptively weighting the nodes, the subsequent personalized model will not perform better than our method. We really appreciate it if the reviewer can suggest us additional baseline methods so that we will include them in the next submission.
>
> **Q: The experiments are conducted only on image classification datasets. I suggest the authors consider using some recommendation datasets (e.g., Movielens) since the user data in these datasets are naturally heterogeneous (and more realistic than simulated data).**
>
> A: We believe the setting of cross-device federated learning is more suitable for training a recommendation system because the goal is to ensure each user will receive good recommendations on his or her personal device. However, as we mentioned above, we are motivated by the applications under the cross-silo setting where each node is an institute. We are not sure how to split a recommendation dataset (e.g., Movielens) into multiple nodes to set up the environment for testing. For example, should we split users or movies or both? Due to the limit of time, we are not able to add the exact experiment requested by the reviewer in the revision, but we will consider it an important future work to extend our method for training recommendation systems. In the revision we submitted, we have added some experiment on a dataset that is not for image classification. In particular, we consider a linear logistic regression model for multi-class classification on the Covtype dataset where the goal is to predict forest cover type using cartographic information. Please see this new result in Section G.1.5 in Appendix.
>
> **Q: The real communication costs of the proposed method and existing methods are not reported. It is suggested to include the results to confirm that the proposed method is "communication-efficient".**
>
> A: In fact, we do show communication costs in our results as the number of synchronizations in Figure 1 and Figure 6. Here, in one synchronization (also called one round of communication), all nodes stop their local computation and exchange models between the center and the participating nodes. This is often the most time consuming step of a federated learning method. Hence, the number of synchronizations measures the communication costs. From Figure 1 and 6, we can see that our bilevel method achieves a higher accuracy using fewer synchronizations, showing that our method is more communication-efficient.

---

### Official Review · Reviewer_Caqk · 2022-10-23

**Confidence:** 3
**Clarity, Quality, Novelty And Reproducibility:** The quality, clarity and originality …
**Correctness:** 3
**Technical Novelty And Significance:** 2
**Empirical Novelty And Significance:** 2
**Recommendation:** 6

**Strength And Weaknesses:**

Pros:

1. Motivation of this paper is reasonable to me and the weighted node formulation seems new in the literature of federated learning although it is not new in other multi-task or device applications such as meta-learning or life-long learning (Chen et al., 2021). In addition, the bilevel optimization based formulation provides a principled way to improve the generalization.

2. Generalization performance of the proposed weighted node is provided, with several modifications of the analysis in Chen et al., 2021. Some new characterizations such as identical neighbors and error bound conditions are also included.

3. Convergence analysis is also provided under local-svrg and some accelerated bilevel methods as in Ghadimi & Wang, 2018.

Cons:

1. There are too many notations and it is a little hard to find the definition among words. I recommend having a brief summary of all notations, which is helpful for reading both the article and the proof.

2. The discussion of communication rounds is not conspicuous although the author strengthened the algorithm is communication-efficient as this is important in FL. I recommend having a table or a figure to show the communication rounds of different algorithms so that readers can intuitively get this.

3. For the generalization analysis, Assumption 2 is strong although some simple examples and Assumption 2’ are provided. This assumption requires $p_0$ is identical to at least one distribution in $p_1,…,p_K$. First, for the practical consideration,  the training-validation separation trick does not hold here, and the generalization may be affected as well. In addition, in terms of the technical novelties, the improved analysis by the error bound condition and the  identical neighbors (Assumption 2’) seems to be not that significant given the framework in Chen et al., 2021, and no tightness results are provided.

4. The algorithm seems not to be new enough. For example, it seems to be a combination of local-SVRG, accelerated bilevel update in  Ghadimi & Wang 2018, and FedAvg types of aggregations. The algorithm also looks complicated with many hyperparameters, and its application may have some difficulty in more complex ML applications. I wonder if the acceleration step truly provides practical acceleration? Could you please include such an illustration in the experiments?

5. Some important related works are missing:

Distributed bilevel optimization:

1. Ataee Tarzanagh, Davoud, et al. "FEDNEST: Federated Bilevel, Minimax, and Compositional Optimization." arXiv e-prints (2022): arXiv-2205.
2. Chen, Xuxing, Minhui Huang, and Shiqian Ma. "Decentralized bilevel optimization." arXiv preprint arXiv:2206.05670 (2022).

3. Li, Junyi, Feihu Huang, and Heng Huang. "Local Stochastic Bilevel Optimization with Momentum-Based Variance Reduction." arXiv preprint arXiv:2205.01608 (2022).

Bilevel optimization and analysis:

4. Ji, Kaiyi, Junjie Yang, and Yingbin Liang. "Bilevel optimization: Convergence analysis and enhanced design." International Conference on Machine Learning. PMLR, 2021.

5. Grazzi, Riccardo, et al. "On the iteration complexity of hypergradient computation." International Conference on Machine Learning. PMLR, 2020.

**Summary Of The Paper:**

This paper focuses on Federated learning (FL) with heterogeneous data distributions. In this case, a globally shared model may not achieve a good local generalization performance. So, the author put forward a method that each node can still exploit global data through FL but, at the same time identify and collaborate only with the nodes whose data distributions are similar or identical to its local distribution. Furthermore, the author mentioned, it is also a bilevel optimization (BO) problem where the inner problem is federated learning with weighted nodes, and the outer problem optimizes the weights. In addition, the theorem also shows the bound on the statistical distance, dominated by the $O(1/\sqrt{N_{valid}})$, which is the same as the generalization bound achieved by leaning locally. Finally, the algorithm in this article is well-consistent with the analysis of BO since it solves the inner problem before the outer problem. As a result, the algorithm performs better than the baseline when the data is heterogeneously distributed where higher test accuracy can be reached.

**Summary Of The Review:**

Overall, this paper provides an interesting bilevel formulation, which is shown to improve the local generalization performance. The analysis involves several new treatments. However, I have some concerns regarding analysis, related work and writing. I think this is a borderline work, and my evaluation will be adjusted based on the other reviewers’ comments and the authors’ response.

---

> ### Author Response · Authors · 2022-11-18
> **Response to Reviewer Caqk**
>
> Thank you so much for your time and comments. Here are the answers to the cons.
>
> 1: Thanks for this point. We will add a summary of notions at the beginning of the appendix in the next version.
>
> 2: We agree that a table will make the comparison on communication rounds more clear. We will add it under Remark 1 in the next version.
>
> 3: Could the reviewer please explain what he/she means by ``the training-validation separation trick does not hold here, and the generalization may be affected as well'', so that we can better answer your questions? Maybe Assumption 2 or 2' can be stated in a weaker form, but something like Assumption 2 is necessary to ensure problem $\widehat P$ is relevant to federated learning. Without Assumption 2, we don't even know if data from any of $p_{1},\dots,p_{K}$ will help learning from $p_{0}$ at all. Then it may happen that $p_{1},\dots,p_{K}$ are useless and the best solution is just learning locally, which can happen in practice but is not what we consider in a federated learning setting.
>
> We totally agree that our results are mostly built on Chen et al. (2021), but our results do bring new insights. In particular, if we only use the results by Chen et al. (2021) or our Theorem 1 (a small improvement of Chen et al. (2021)), the bilevel method $\widehat P$ might not perform better than local training because the theory  (see Theorem 1 and (10)) indicates that we may also need lots of data to learn the weight accurately. The main challenge is how to improve the dependence of our sample complexity on $n_{\text{valid}}$ to make it better than  $O(1/\sqrt{n_{\text{valid}}})$. Otherwise, the bound is asymptotically the same as just training locally. Fortunately, we figure out that this is possible if the error bound condition holds and the statistical distance $G$ between $p_0$ and $p_k$’s is small (but not necessarily zero). See Theorem 2 and Corollary 1. This is a new finding that can explain why the bilevel method performs better than local training emprically while Chen et al. (2021) cannot explain.
>
> Regarding the tightness of Theorem 1, I believe the $O(1/\sqrt{n_{\text{valid}}})$ term is the bottle neck and it is not improvable unless additional conditions are made (e.g., the error bound condition we made). We do not know if the bounds in Theorem 2 and Corollary 1 are tight or not, which is one of our directions to explore.
>
> 4: We agree with the reviewer that the algorithm is just a combination of existing methods. The main contribution is a new strategy of doing federated learning when not all nodes are helpful, which is fundamentally different from all existing federated learning frameworks. The theoretical contribution of this paper is the generalization bound, especially, Theorem 2 and Corollary 1. In theory, if $\widehat F(w)$ in $\widehat P$ is convex, the acceleration step in Algorithm 2 can reduce the number of communication rounds for solving $\widehat P$ from $O(1/\epsilon)$ and $O(1/\epsilon^{0.5})$. When $\widehat P$ is non-convex, the acceleration step does not reduce the communication rounds (does not increase neither). As a result, in the non-convex case, people just use the non-accelerated gradient method described in Algorithm 3 in our appendix. Since the instances we tested in our numerical experiments are all non-convex, we do implement Algorithm 3 instead of the accelerated algorithm (Algorithm 2).
>
> 5: Thank you so much for pointing out the  related works we missed. We actually have already cited 1 and 3 in the initial submission. We have also cited other papers you suggested in the revision we just submitted.

---

> > ### Comment · Reviewer_Caqk · 2022-12-01
> > **Thanks for the response**
> >
> > I thank the authors for the detailed response. Most of my concerns are addressed. I increase my rating accordingly.
> >
> > Best,
> > Reviewer

---

### Official Review · Reviewer_F6ZM · 2022-10-25

**Confidence:** 4
**Correctness:** 3
**Technical Novelty And Significance:** 2
**Empirical Novelty And Significance:** 2
**Recommendation:** 3

**Clarity, Quality, Novelty And Reproducibility:**

The paper is well written, with good clarity.
Even though they are the first to adopt a bi-level formulation for FL, this approach is quite straightforward and does not present enough novelty. Besides, some major limitations of bilevel formulations, such as expensive inner-loop evaluation, are not addressed in this paper. The increase in global communication rounds also introduces more difficulty in applying this method.


**Strength And Weaknesses:**

Strength:

The paper seems to be the first one to formulate node reweighting for FL into a bi-level optimization problem, and theoretical results are given to support the superiority of training a model locally and to federated learning with static and evenly distributed weights.

Weaknesses:

1. It is not surprising that FL with properly learnt reweighting performs better than local training and evenly distributed weights, which is the main theoretical contribution of this paper.
2. The proposed bi-level reweighting method does not seem to be applicable in practice. The algorithm needs S X T global communication rounds, just to acquire the properly learnt node reweighting. This is obviously not acceptable, since global communication is often the main bottleneck in the application of FL. In addition, in the inner loop of Local-SVRG, the model seems to be re-initialized every time, this does not seem to match the real scenario, since we cannot ask the clients to re-initialize their models now and then.
3. It seems that the weights are assigned to each specific client node, will it be problematic when the participation ratio is not 1 when running FL? This is often the case in practice, and this paper does not discuss about it.
4. The bilevel method needs an in-distribution validation set,  and the size of such a dataset will affect the performance. An ablation study is required to clarify this.
5. The algorithm is built based on Local-SVRG, which should serve as a baseline method in the experiment, so that we can see how much performance gain the reweighting actually brings.



**Summary Of The Paper:**

This paper introduces a node reweighting strategy for federated learning based on bilevel optimization. Specifically, this the proposed adaptive reweight method learns the weight assigned to each node by considering the “weighted aggregation” on the server as the inner problem, and minimizes the empirical risk on an in-distribution validation dataset. Theoretical results are given to verify the effectiveness of the bilevel method.

**Summary Of The Review:**

This paper has certain contribution, but I do not think it is enough for getting accepted by ICLR. Some major limitations need to be addressed for the applicability of the proposed method.

---

> ### Author Response · Authors · 2022-11-18
> **Response to Reviewer F6ZM**
>
> Thank you so much for your time and comments. Here are the answers to the weaknesses.
>
> 1: It may not be surprising from a practical perspective, but it is still good to establish some theoretical guarantee. In fact, from the theoretical perspective, learning the weights might not do better than only learning locally because the theory (see Theorem 1 and (10)) indicates that we may also need lots of data to learn the weight accurately.  The main challenge we had is how to improve the dependence of our sample complexity on $n_\text{valid}$ to make it better than $O(1/\sqrt{n_\text{valid}})$. Otherwise, the bound is asymptotically the same as just training locally. Fortunately, we figure out that this is possible if the statistical distance $G$ between $p_0$ and $p_k$’s is small (but not necessarily zero).
>
> 2: We assume $S$ and $T$ the reviewer wrote here are $S$ and $T_s$ in Algorithm 2 and 3. First, we want to clarify that, after the required number of global communication rounds, the algorithm returns both the node reweighting and the model trained using the node reweighting. Our method is not a two-stage process where the weights are learnt first. Instead, each communication round updates the model also instead of just the weights. Second, the number of global communication rounds is not exactly $S\times T$ but $\sum_{s=0}^{S-1} T_s$ with $T_s$ increasing in $s$. Whether the number is large or not depends on how $\sum_{s=0}^{S-1} T_s$ increases as $\epsilon$ decreases. In fact, the total number of rounds is not too large. For the convex case, we need only $O(1/\epsilon^{0.5})$ rounds, which is very small in federated learning literature. For the non-convex case, we need only $O(1/\epsilon^2)$ rounds, which are lower than existing methods (see the comparison in Remark 1). Of course, we need fewer communication rounds because our problem is a finite-sum (deterministic) problem instead of a stochastic problem. Lastly, we apologize for not being clear enough on how each call of Local-SVRG is initialized in Algorithm 2 and 3. In fact, we always used the output model from the last call to initialize the inner-loop of Local-SVRG in the next call. This does not affect any theoretical results but is more practical as the reviewer said.
>
> 3: Thank you so much for pointing this out!  We actually focus on the cross-silo scenario instead of the cross-device scenario. In the cross-silo scenario, the center and nodes are a few institutes such as hospitals whose participation are almost always guaranteed. However, we should definitely include the discussion on the issue of participation ratio in the next submission as the reviewer suggests. Our method essentially utilizes an existing federated learning method for solving the weighted loss minimization subproblem in (1) for fixed weights. Since we assume full participation of nodes, we use Local-SVRG by Gorbunov et al. (2021) which requires full participation also and has the lowest communication complexity. If we have only partial participation, we just need to replace Local-SVRG by a different existing federated learning method for solving (1). One potential choice is FedAvg in https://arxiv.org/pdf/1907.02189.pdf which provides theoretical communication complexity analysis with partial participation. We will numerically evaluate the testing performance of $\widehat P$ with partial participation in the next revision. Please note that the main focus of this paper is the theoretical generalization bounds in Section 3 which are based on the optimal solution of $\widehat P$, so these results are not undermined as long as $\widehat P$ can be still solved with partial participation.
>
> 4: We agree with the reviewer. We choose Setting 2 in our experiments and run the experiments using validation sets of different sizes and watch how the performances are affected. For FashionMNIST, MNIST and Cifar-10, we run with sizes 100, 200, 500 and 1000. For ImageNet, we run with sizes 1000, 1200, 1500 and 2000. Please see Section F.4 in the revision we submitted. Overall, the results show that a larger validation set improves the testing performance in general. As long as the size of validation set is not too small (e.g. 100), the testing performance of our method is stable with an average change less than 5\% when the size of validation set changes. Due to the limit of time, we only try four different sizes and we will test using more sizes in the next revision.
>
> 5: In fact, FedAvg in all of our experiments is essentially Local-SVRG as we stated on page 8. Based on our understanding, FedAvg is a general approach using a local solver (mainly SGD in literature but not necessarily) to minimize the local loss and periodically averages the solutions through communication. We use SVRG as the local solver in FedAvg, so the resulting algorithm is just Local-SVRG. We apologize for the confusion of notation and we will label FedAvg as Local-SVRG in all tables and figures in the next revision.

---

### Official Review · Reviewer_gCZG · 2022-10-25

**Confidence:** 4
**Clarity, Quality, Novelty And Reproducibility:** The paper is clearly written, novelty…
**Correctness:** 4
**Technical Novelty And Significance:** 2
**Empirical Novelty And Significance:** 2
**Recommendation:** 3

**Strength And Weaknesses:**

*Strength*
The problem is interesting, paper is well written withe extensive experiments.

*Weakness*

- The contribution is limited and incremental.
- Once the problem is formulated in the form of P*, I cannot see any additional challenge to solve it. There are multiple algorithms already existing in the literature which can directly solve this problem.
- It is limited to consider the lower level problem as strongly convex, because even the simplest FL problem where we train NN would be non-convex. For instance, all the experiments are definitely considering non-convex lower level, therefore theoretical results are not consistent with the experiments.
- Assuming the lower level to be strongly convex makes the setting quite straightforward to analyze and provide guarantees. It would be nice to specifically discuss the additional challenges, theoretically, authors had to address to do the analysis.
-

**Summary Of The Paper:**

The authors have considered the problem of designing weights for federated learning problem via bilevel optimization. The problem is interesting and analyzed theoretically with extensive experiments. But there are certain limitations as provided next.

**Summary Of The Review:**

The problem is interesting, but the contribution is incremental.

---

> ### Author Response · Authors · 2022-11-18
> **Response to Reviewer gCZG**
>
> Thank you so much for your time and comments. The following are our responses to your questions and concerns.
>
> **Q: Once the problem is formulated in the form of $P^{*}$, I cannot see any additional challenge to solve it. There are multiple algorithms already existing in the literature which can directly solve this problem.**
>
> A: Could we confirm whether the reviewer meant $P^*$ or $\hat{P}$ in your first sentence? If you meant $P^*$, that is not what we want to solve. Problem $P^*$ is only needed in the assumption and the theoretical proofs. We want to solve $\hat{P}$ actually. We agree that $\hat{P}$ can be solved easily under the single-machine setting and there have been many papers on bilevel optimization under the single-machine setting. However, there are much fewer works under the distributed setting. That said, we acknowledge that our optimization algorithm is not very new as it just combines Gorbunov et al. (2021) and Ghadimi & Wang (2018). That's not the main contribution of this paper. The main contribution is a new strategy of doing federated learning when not all nodes are helpful, which is fundamentally different from all existing federated learning frameworks. The theoretical contribution of this paper is the generalization bound, especially, Theorem 2. The optimization algorithms and their convergence results are included mainly for completeness. We could have removed the optimization sections completely. However, without giving an optimization algorithm, readers who are not familiar with distributed optimization might not know how to solve  $\hat{P}$ with distributed data, and we will also need to introduce the optimization algorithm to describe the numerical experiments in details.
>
> **Q: It is limited to consider the lower level problem as strongly convex, because even the simplest FL problem where we train NN would be non-convex. For instance, all the experiments are definitely considering non-convex lower level, therefore theoretical results are not consistent with the experiments.**
>
> A: To solve a bilevel optimization problem, we need to assume the strong convexity of lower problem to ensure  $\widehat\theta(w)$ is uniquely defined for a given $w$ and $\widehat F(w)$ is differentiable in $w$. This is the assumption made in almost all recent papers on bilevel optimization models in machine learning under both the single-machine and federated settings. It is hard to find one paper without this assumption. The only one we are aware of is https://arxiv.org/pdf/2203.01123.pdf which still assumes convexity (but not strong convexity).
>
> That said, we agree with the reviewer that there exists a gap between theoretical results and experiments. In the revision, we have added some experiment on a linear logistic regression model for multi-class classification. The lower level problem with this model is strongly convex as the dataset, Covtype,  we used has more rows than columns. Due to the limit of time, we only complete the test in one dataset for the convex case. Please see this new result in Section G.1.5 in Appendix.
>
> We would like to include more numerical experiments on the convex case for the next submission. It will be an award-winning paper if we can do bilevel optimization without convexity assumption on the lower level. We are ambitious to explore that in the future.
>
> **Q: Assuming the lower level to be strongly convex makes the setting quite straightforward to analyze and provide guarantees. It would be nice to specifically discuss the additional challenges, theoretically, authors had to address to do the analysis.**
>
> A: Thank you for this great suggestion. We should definitely emphasize the challenges we had on analyzing the generalization bounds rather than the optimization bounds. The main challenge we had is how to improve the dependence of our sample complexity on $n_{\text{valid}}$ to make it better than  $O(1/\sqrt{n_{\text{valid}}})$. Otherwise, the bound is asymptotically the same as just training locally. Fortunately, we figure out that this is possible if the statistical distance $G$ between $p_0$ and $p_k$’s is small (but not necessarily zero). Please see Theorem 2 and Corollary 1.

---

### Decision · Program_Chairs · 2023-01-20

**Decision:**

Reject

**Justification For Why Not Higher Score:**

The contribution and novel are not sufficient.

**Justification For Why Not Lower Score:**

N/A

**Metareview: Summary, Strengths And Weaknesses:**

This paper presents a bilevel optimization-based federated learning method that can assign different nodes different weights to optimize the performance on a validation dataset. The authors present thorough theoretical analysis of the problem and verify that learning dynamic weights is better than using evenly distributed weights. Experimental analysis also shows the superiority of the proposed method over several baseline methods.

There remains concerns on the proposed approach being straightforward, and the contribution / novelty may not be significant enough for ICLR. We thus unfortunately recommend rejection.